# Improving Explicit Dynamic Gaussian Splatting Optimization via Update Mixture

Renjie Ding [1]   Yaonan Wang [1]   Min Liu [1]   Jialin Zhu [2]   Jiazheng Wang [1]   Jiahao Zhao [1]
Xiao Tan [1]   Feixiang He [3]   Xiang Chen [1]

## Abstract

3D Gaussian Splatting (3DGS) enables real-time, high-fidelity view synthesis via explicit scene representations and has recently been extended to dynamic scene modeling. Despite their excellent rendering quality and interpretability, we find that explicit Dynamic GS often exhibits generalization degradation in scenes with large motion. Motivated by generalization behavior in deep neural optimization and the characteristics of Gaussian primitive optimization, we propose an update mixture strategy. This work focuses on two representative open-source explicit Dynamic GS pipelines and our approach consists of three components: (i) a space–time dependent Strictly Sparse Update with additional regularization to stabilize adaptive updates; (ii) a constant-corrected adaptive algorithm that alleviates over-scaling of primitive gradients and yields a stable mixture of adaptive and non-adaptive steps; and (iii) attributes mixing via Stochastic Attribute Averaging to mitigate frame-preference under motion disturbances. Experiments show consistent improvements and reduced generalization issues, highlighting the role of non-adaptive updates and the impact of frame-preference in explicit Dynamic GS optimization.

## 1. Introduction

Modeling dynamic scenes from 2D images and rendering photorealistic novel views in real time are central problems in computer vision and graphics, with broad applications in augmented reality, virtual reality, and digital twins. Despite major advances driven by Neural Radiance

[1]National Engineering Research Center of Robot Visual Perception and Control Technology, School of Artificial Intelligence and Robitics, Hunan University [2]Baidu Inc. [3]Central South University. Correspondence to: Yaonan Wang <yaonan@hnu.edu.cn>.

*Proceedings of the $43^{rd}$ International Conference on Machine Learning*, Seoul, South Korea. PMLR 306, 2026. Copyright 2026 by the author(s).

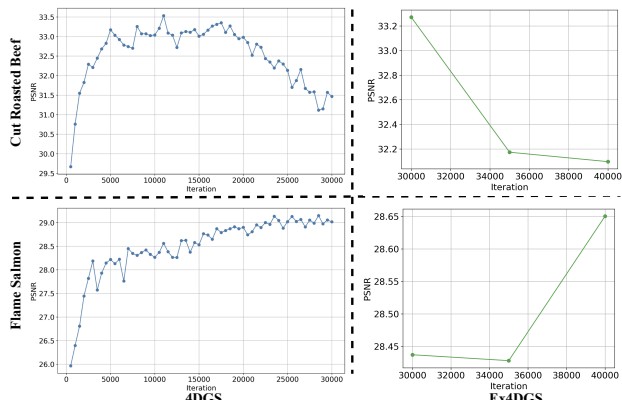

*Figure 1.* We evaluate the performance of different methods on the test set at different training iterations. Each row corresponds to one scene, i.e., Cut Roasted Beef and Flame Salmon from N3DV (Li et al., 2022), while each column corresponds to one method, i.e., 4DGS (Yang et al., 2024) and Ex4DGS (Lee et al., 2024). As Ex4DGS adopts a progressive training scheme, we only evaluate the endmost iterations. We observe that scenes with larger motion are more prone to generalization issues, where test-set performance degrades in the later stages of training. A visualization of the motion amplitude is provided in Appendix Section C.1.

Fields (NeRF) (Mildenhall et al., 2021; Sun et al., 2022; Müller et al., 2022), NeRF-based rendering often incurs non-trivial latency. More recently, 3D Gaussian Splatting (3DGS) (Kerbl et al., 2023) has provide an efficient alternative by representing a scene as explicit Gaussian primitives that can be efficiently rasterized, enabling real-time photorealistic rendering. Owing to this efficiency, 3DGS has gained substantial traction for static scenes and has been widely adopted across diverse applications (Han et al., 2024; Cheng et al., 2024). However, extending 3DGS to dynamic scenes remains comparatively underexplored.

Dynamic Gaussian Splatting methods—hereafter referred to as **Dynamic GS**—can be broadly categorized into two paradigms. The first, inspired by NeRF, relies on MLPs to model the temporal evolution of primitive attributes (Wu et al., 2024). Introducing such an implicit function weakens the explicit interpretability of primitive motion and leads to a more complex optimization system. The second paradigm explicitly extends vanilla 3DGS by introducing an addi-

tional temporal dimension to model time-varying Gaussian primitives (Yang et al., 2024; Lee et al., 2024), yielding a more intuitive and interpretable representation of primitive motion. In practice, explicit Dynamic GS methods also tend to achieve better reconstruction quality.

However, as shown in Figure 1, we observe that explicit Dynamic GS suffers from pronounced generalization degradation in large-motion scenes. Specifically, reconstruction quality is not guaranteed to improve during training; instead, it can deteriorate substantially, with the test-set PSNR value sometimes dropping sharply toward the end of training. To rule out incidental effects, we examine two representative open-source explicit Dynamic GS methods, 4DGS (Yang et al., 2024) and Ex4DGS (Lee et al., 2024), which adopt markedly different pipelines. This phenomenon is consistently observed in both methods.

Generalization across test viewpoints is typically not a major concern in dense-view 3DGS. In Dynamic GS, however, generalization issues commonly emerge in the later stages of training and become more pronounced in scenes with larger motion. This suggests that once Dynamic GS fits erroneous information induced by strong motion disturbances, it struggles to correct such errors and may become trapped in a suboptimal optimization regime.

Although 4DGS and Ex4DGS differ substantially in their pipeline designs, both rely on a similar optimization paradigm: adaptive gradient-based optimization, with Adam used in 4DGS and RAdam (Liu et al., 2020) used in Ex4DGS. The generalization behavior of adaptive optimizers has been extensively studied in deep neural network (DNN) training. Existing remedies typically fall into two lines of work: strengthening regularization and mitigating non-uniform gradient scaling by reducing adaptivity, for example, by introducing non-adaptive updates. Building on the optimization characteristics of Gaussian splatting and its structured gradient patterns, this work improves Dynamic GS optimization through an update-mixture strategy.

- To construct reliable adaptive updates, we first develop a Strictly Sparse Update that restricts update steps to accessible attributes, and then introduce additional regularization to stabilize optimization.

- Inspired by the benefits of non-adaptive updates and the characteristics of primitive gradients, we propose a constant correction for adaptive optimization, which turns the original optimization process into a mixture of adaptive and non-adaptive steps.

- Considering the frame-preference caused by the motion disturbance, we introduce the stochastic attribute averaging to keep the reliable attributes update.

- This work highlights the impact of non-adaptive up-

dates and frame-preference effects in Dynamic GS optimization. With the proposed strategy, generalization degradation is alleviated and better performance is achieved.

## 2. Related Work

**3DGS** has become a competitive representation for static-scene rendering and has been adopted in diverse downstream applications, including scene memory (Zou et al., 2025), robotics and autonomous systems (Matsuki et al., 2024; Yang et al., 2025), 3D reconstruction (Yu et al., 2024), and 3D generation (Xu et al.; Deng et al., 2025). Subsequent work has extended 3DGS along multiple directions, such as primitive redesign (Huang et al., 2024; Yu et al., 2024), improved densification strategies (Rota Bulò et al., 2024; Fan et al., 2024), and accelerated rendering (Durvasula et al., 2023). In parallel, recent studies have examined 3DGS optimization: one line of work formulates primitives as stochastic samples from an underlying distribution and integrates stochastic gradient Langevin dynamics (Kheradmand et al., 2024); others analyze sparse-view generalization by characterizing gradient decay and appearance artifacts (Park et al., 2025; Chen et al., 2026); and another study reports that many primitives remain near saddle regions during training (Wang et al., 2025b).

**Dynamic Gaussian Splatting** can be broadly categorized into two groups. The first group represents dynamic scenes by jointly leveraging Gaussian primitives and MLPs. Within this group, some methods directly employ MLPs to model the deformation of primitive attributes (Wu et al., 2024), while others further refine dynamic attributes through MLPs (Li et al., 2024; Gao et al., 2025). However, introducing MLPs weakens the explicit structural representation inherent in 3DGS, thereby limiting the ability to explicitly exploit motion-induced correlations. Moreover, the coupling between Gaussian primitives and neural networks may lead to more complex optimization dynamics. In contrast, the second group models motion explicitly by extending the original 3DGS formulation without relying on MLPs, either by introducing an additional temporal dimension and computing marginal distributions (Yang et al., 2024), or by interpolating between 3DGS keyframes to model trajectories in 3D space (Lee et al., 2024).

**Weight Averaging** originates from SGD-based convex optimization, where averaging iterates under constant or cyclical learning rates achieves asymptotic statistical optimality for certain locally quadratic objectives (Ruppert, 1988; Polyak & Juditsky, 1992). This idea was later introduced into DNN training as Stochastic Weight Averaging (SWA) (Izmailov et al., 2018), which improves generalization by finding wider optima. SWA has since been adopted in language models (Lu et al., 2022), low-precision training (Yang et al.,

2019), and other settings. However, SWA is sensitive to the choice of the averaging start point, and adaptive variants inspired by early stopping have been proposed to alleviate this issue (Demir et al., 2024). To the best of our knowledge, weight averaging has not yet been explored in Gaussian Splatting.

**Generalization of Adam.** Adaptive gradient methods, such as Adam, have become a standard choice for training DNNs (Radford et al., 2015). However, empirical studies have shown that Adam often generalizes worse than Stochastic Gradient Descent (SGD) with momentum across a wide range of tasks (Wilson et al., 2017). Subsequent work has sought to understand the underlying causes of this generalization gap from multiple perspectives, including training dynamics (Cohen et al., 2022), statistical properties (Jiang et al., 2023), and sharp minima (Granziol, 2020; Zhou et al., 2020; Xie et al., 2022). From a theoretical standpoint, backward error analysis provides important insights, revealing implicit regularization in SGD (Miyagawa, 2022; Smith et al.) and, in contrast, anti-regularization cases in adaptive methods such as Adam (Cattaneo et al., 2024; Cattaneo & Shigida, 2025). Despite their potential generalization disadvantage, adaptive methods remain attractive in practice because of their rapid initial progress, relative insensitivity to hyperparameter tuning, and robustness to ill-conditioned or poorly scaled problems. Motivated by this trade-off, several works have proposed hybrid strategies, such as switching from Adam to SGD during later training stages (Keskar & Socher, 2017). Alternatively, explicit regularization schemes have been introduced for adaptive optimizers, most notably AdamW (Loshchilov & Hutter), which decouples weight decay from the adaptive gradient update.

However, due to the distinctive pipeline of 3DGS, as well as the physical semantics and attribute-specific optimization characteristics of Gaussian primitives (Kerbl et al., 2023; Jung et al., 2024; Wang et al., 2025a;b; Mo et al., 2025), generalization techniques developed for conventional DNNs are not directly applicable. As a result, existing approaches either remain incompatible with 3DGS or still rely on Adam-based densification (Höllein et al., 2025), and become even less suitable for the more complex Dynamic GS setting.

## 3. Preliminary

**Gaussian Splatting.** Typically, Gaussian primitives are employed to model static scenes. Each primitive is parameterized by a set of attributes, including position $\boldsymbol{\mu}$, covariance $\boldsymbol{\Sigma}$, opacity $o$ and color $\boldsymbol{c}$. Given a viewpoint $v$ and the associated camera transformation, each primitive is transformed into the camera coordinate and projected onto the image plane, yielding a 2D Gaussian footprint $\mathcal{G}_i(\boldsymbol{p}; \boldsymbol{\mu}_i^v, \boldsymbol{\Sigma}_i^v)$ at pixel $p$. The pixel color is then rendered through $\alpha$-blending:

$$C(\boldsymbol{p}) = \sum\nolimits_{i=1}^{N_p^v} \boldsymbol{c}_i^v o_i \mathcal{G}_i(\boldsymbol{p}; \boldsymbol{\mu}_i^v, \boldsymbol{\Sigma}_i^v) T_i,$$
$$\text{where} \quad T_i = \prod\nolimits_{j}^{i-1}(1 - o_j \mathcal{G}_j(\boldsymbol{p}; \boldsymbol{\mu}_j^v, \boldsymbol{\Sigma}_j^v)) \tag{1}$$

Here, $N_p^v$ denotes the number of primitives visible at pixel $\boldsymbol{p}$ from viewpoint $v$. The indices $i$, $j$ refer to primitives ordered by rendering depth. $\boldsymbol{c}^v$, $\boldsymbol{\mu}^v$ and $\boldsymbol{\Sigma}^v$ are the corresponding view-transformed attributes after applying the camera transformation (Kerbl et al., 2023).

To model dynamic scenes, explicit Dynamic GS introduces an additional temporal dimension and extends Eq. 1 to Eq. 2:

$$C(\boldsymbol{p}, t) = \sum\nolimits_{i=1}^{N_p^v} \boldsymbol{c}_i^v o_i(t) \mathcal{G}_i(\boldsymbol{p}; \boldsymbol{\mu}_i^v(t), \boldsymbol{\Sigma}_i^v(t)) T_i \tag{2}$$

where $o_i(t)$ accounts for the appearance and disappearance primitive $i$ and is commonly instantiated as an opacity attribute modulated by a time-dependent probability (Yang et al., 2024; Lee et al., 2024). Meanwhile, $\boldsymbol{\mu}_i(t)$ and $\boldsymbol{\Sigma}_i(t)$ govern the temporal evolution of its geometry. Notably, 4DGS and Ex4DGS instantiate these time-varying attributes in fundamentally different ways.

Without using MLPs, 4DGS (Yang et al., 2024) models $\boldsymbol{\mu}(t)$ as a **continuous** function, which can be simplified as $\boldsymbol{\mu}(t) = \boldsymbol{\mu} + \gamma(t - \mu_t)$, where $\gamma$ and $\mu_t$ denote simplified coefficients. In contrast, Ex4DGS (Lee et al., 2024) represents $\boldsymbol{\mu}(t)$ and $\boldsymbol{\Sigma}(t)$ as **temporal sequences of discrete values**. Given a timestamp $t$, the corresponding temporal index is first computed, after which the target value is obtained by accessing a *subset of entries* in the sequence and interpolating between them.

**Adaptive Update Step.** Adaptive gradient algorithms, such as Adam and RAdam, maintain exponential moving averages of the gradients and squared gradients, denoted by $\boldsymbol{m}$ and $\boldsymbol{v}$ as first- and second-moment estimates, to introduce historical information into the optimization process in Eq. 3. This historical accumulation enables a per-parameter **adaptive update rule**, which applies non-uniform scaling to parameter updates by assigning larger effective step sizes to parameters with small accumulated gradients and smaller step sizes to those with large or high-variance gradients.

$$\boldsymbol{m}_k = (1 - \beta_1)\boldsymbol{m}_{k-1} + \beta_1 \boldsymbol{g}_k$$
$$\boldsymbol{v}_k = (1 - \beta_2)\boldsymbol{v}_{k-1} + \beta_2 \boldsymbol{g}_k^2 \tag{3}$$
$$\boldsymbol{\theta}_{k+1} = \boldsymbol{\theta}_k - \frac{\boldsymbol{\eta}}{\sqrt{\hat{\boldsymbol{v}}_k} + \epsilon} \circ \hat{\boldsymbol{m}}_k$$

Here, $\circ$ denotes element-wise product. $\boldsymbol{\eta}$ is the predefined learning rate for adaptive updates, and $\epsilon$ is a constant introduced for numerical stability. $\boldsymbol{\theta}$ and $\boldsymbol{g}$ denote all trainable parameters and their corresponding gradients, respectively. Parameters refer to optimization variables rather than directly rendered attributes; for instance, opacity is obtained by applying a sigmoid function to its underlying parameter.

# 4. Methods

To improve generalization or optimization behavior in explicit Dynamic GS, we draw on insights from previous studies of DNN optimizationg while accounting for the gradient patterns and optimization characteristics of explicit Dynamic GS. We first establish a reliable adaptive update step in Section 4.1 by restricting updates according to spatial and temporal accessibility and introducing fixed regularization terms during optimization. To attenuate the over-scaling of primitive gradients, we apply a constant correction to the adaptive update step in Section 4.2. With this simple modification, we construct a stable hybrid update strategy that combines adaptive and non-adaptive updates. Furthermore, considering frame-preference effects caused by motion disturbances in frame-by-frame optimization, we incorporate Stochastic Attribute Averaging to maintain reliable attribute updates in Section 4.3.

## 4.1. Reliable Update

**Strictly Sparse Update** Taming-3DGS (Mallick et al., 2024) first introduces sparse updates in Adam to improve the training efficiency of 3DGS. Subsequent work further leverages sparse updates for more stable optimization, reducing sensitivity to logarithmic parameterization (Pateux et al., 2025). Empirical evidence in (Ding et al., 2026) also confirms these stability benefits, and sparse updates lead to more reliable moment estimates. In 3DGS, sparse updates are typically defined solely based on a spatial visibility condition $\mathcal{V}(v)$, which is determined by the viewpoint and obtained via frustum culling (Kerbl et al., 2023). In Dynamic GS, however, primitives evolve over time and may appear or disappear, inducing time-varying accessibility. We therefore introduce an additional temporal condition $\mathcal{T}(t)$, whose entry is zero when the corresponding primitive is inactive at time $t$. Combining $\mathcal{V}(v)$ and $\mathcal{T}(t)$, we define the accessible mask $\mathcal{A}(v, t)$ in Eq. 4 and accumulate update steps only for primitives in $\mathcal{A}(v, t)$.

$$\mathcal{A}(v,t) = \mathcal{V}(v) \circ \mathcal{T}(t), \quad \mathcal{U}(v,t) = \mathrm{diag}(\mathcal{A}) \cdot \mathcal{D}(t) \quad (4)$$

However, as discussed in Section 3, in methods such as Ex4DGS, not all attributes of a primitive are accessible at each time step; the accessibility of each primitive attribute is time-dependent and is encoded by the attribute-accessibility mask $\mathcal{D}(t)$. We therefore restrict updates strictly to accessible attributes, as formalized in Eq. 4. Together with the update-step accumulation constraint, this restriction defines our Strictly Sparse Update (**SSU**).

**Optimization with Regularization** Recent studies suggest that memory in adaptive gradient algorithms can exhibit anti-regularization effects (Cattaneo & Shigida, 2025). In addition, a recent study shows that 3DGS can overly adapt

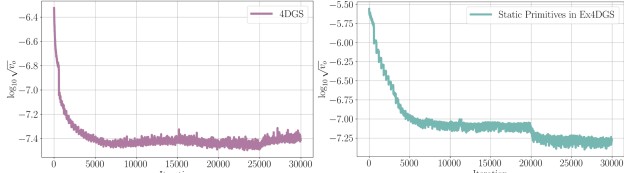

*Figure 2.* $\sqrt{v}$ of the opacity parameter under different methods, shown on a logarithmic scale. Primitives in Ex4DGS (Lee et al., 2024) are grouped into static and dynamic primitives. Here, we report the results for static primitives; additional results are provided in Appendix D.4.

opacity during training, which may induce view-dependent deviations in novel-view rendering (Han et al., 2024). Following (Ding et al., 2026), we therefore introduce opacity regularization and ReState Regularization (**RSR**) as additional training regularization terms. For 4DGS, we disable opacity reset, which has been reported to induce optimization instability (Rota Bulò et al., 2024). We also hypothesize that batch sampling contributes to generalization degradation, as it may lead to underestimated second-moment estimates; therefore, we remove batch sampling in this work. For Ex4DGS, we remove motion regularization on positions because its practical effect is largely negligible, and prior work indicates that improper position updates can drive primitives toward sub-optima (Jung et al., 2024; Mo et al., 2025).

## 4.2. Mix-Adaptive Update Step

Poor generalization is often observed in large-motion scenes. Object motion requires primitives to fit time-varying attributes and spatial relations across frames, inherently inducing substantial inter-frame variability in the gradients of the same primitive. Meanwhile, the non-uniform scaling in adaptive updates is widely recognized to introduce an implicit bias, potentially over-emphasizing certain parameter directions inappropriately (Wilson et al., 2017). Revisiting the gradient characteristics in Dynamic GS, we find a structured disparity in primitive-wise gradient magnitudes; when coupled with non-uniform scaling, this disparity can steer primitives toward suboptimal optimization outcomes.

**Scale of Gradients** To better understand the role of non-uniform scaling in adaptive updates, we first analyze the gradient scales of primitives. Unlike DNNs, which consist of a fixed set of weights, Dynamic GS is composed of time-varying groups of primitives; consequently, gradient scales in Dynamic GS depend on primitive parameterization, spatial relationships, and temporal evolution. Starting from Eq. 1, we derive the per-pixel projection gradient in Eq. 5, where the temporal dimension is omitted for clarity. In contrast, the key distinction in Eq. 2 is that primitive accessibility varies across time steps. Three primary factors govern the gradient scale: (i) the loss gradient $\nabla \ell$, which

typically decreases as training progresses; (ii) opacity $o$, which depends on primitive design—for example, Ex4DGS employs two types of primitives with distinct opacity formulations, leading to different gradient magnitudes; and (iii) the transmittance term $T$, which is determined by spatial relationships. For a given viewpoint, primitives with later rendering orders receive smaller transmittance values, so primitives rendered later, often corresponding to background regions, tend to exhibit smaller gradients. These primitives also affect foreground primitives through $\sum_{j=i+1}^{N} \boldsymbol{c}_j^v \frac{o_j \mathcal{G}_j T_j}{T_{i+1}}$ in Eq. 5. When temporal dynamics are considered, changes in primitive attributes and states, such as appearance or disappearance, further modulate gradient scales.

$$\frac{\partial \ell}{\partial \mathcal{G}_i} = \nabla \ell \cdot (\boldsymbol{c}_i^v - \sum_{j=i+1}^{N} \boldsymbol{c}_j^v \frac{o_j \mathcal{G}_j T_j}{T_{i+1}}) o_i T_i \quad (5)$$

Prior work shows that switching from adaptive methods to SGD in the later stages of training can improve generalization and help escape saddle points (Keskar & Socher, 2017); related saddle-point issues have also been discussed for 3DGS under adaptive optimizers (Wang et al., 2025b). However, the same switching strategy is not directly applicable to Dynamic GS. When all primitives are treated as a single parameter group, the averaged second-moment estimate quickly stabilizes at a nearly fixed order of magnitude, as shown in Figure 2, while the per-primitive second-moment estimates remain highly heterogeneous due to the structured gradient-scale disparity in Eq. 5. In particular, background primitives typically exhibit smaller gradients and are therefore disproportionately amplified by adaptive scaling; their updates can further propagate to foreground primitives through rendering coupling. This naturally raises the question of how to transition adaptive updates toward non-adaptive ones, such as SGD.

**From Adaptive to Non-Adaptive** A simple way to achieve this objective is to limit the amplification of small gradients in adaptive updates. Revisiting Eq. 3, adaptive methods typically regulate the update magnitude using $\sqrt{\hat{v}}$, under the implicit assumption that $\epsilon$ is sufficiently small. When $\epsilon$ is set to a relatively large value, i.e., $\epsilon \gg \sqrt{\hat{v}}$, the adaptive update effectively reduces to a non-adaptive update. Moreover, prior work has proved that attributes within the same primitive have similar magnitudes, except for position (Mo et al., 2025), and we empirically observe that $\sqrt{\hat{v}}$ rapidly stabilizes at a nearly fixed order of magnitude, as shown in Figure 2. Based on these observations, selecting an appropriate fixed $\epsilon$ naturally yields a mixture of adaptive and non-adaptive update steps.

$$\boldsymbol{\theta}_{k+1} = \begin{cases} \boldsymbol{\theta}_k - \frac{\boldsymbol{\eta}}{\epsilon} \circ \hat{\boldsymbol{m}}_k & \epsilon \gg \sqrt{\hat{v}_k} \\ \boldsymbol{\theta}_k - \frac{\boldsymbol{\eta}}{\sqrt{\hat{v}_k}+\epsilon} \circ \hat{\boldsymbol{m}}_k & \text{otherwise} \end{cases} \quad (6)$$

**How to Estimate** $\epsilon_e$  We find that a reliable estimate of the eventual order of magnitude of $\sqrt{\hat{v}}$ can be obtained early in training, as shown in Figure 2, and use this estimate as a reference. This magnitude is largely determined by factors such as primitive design and image scale, which motivates tuning $\epsilon$ across methods and datasets. For Ex4DGS, which employs two primitive types, separate estimates are required for each type. Therefore, for each primitive group, we set $\epsilon$ to $0.1\times$ the estimated initial magnitude for opacity parameters with an additional probability scale, and to $1\times$ the estimated initial magnitude for opacity parameters without an additional probability scale. Moreover, for scenes with small motion, a smaller $\epsilon$ is typically preferable: in these cases, the adaptive rescaling of small gradients already promotes effective optimization, whereas an overly large $\epsilon$ can weaken adaptivity and lead to insufficient learning. We denote the resulting estimate by $\epsilon_e$. A detailed example is provided in Section D.4.

### 4.3. Stochastic Attribute Averaging

Motion disturbances in the frame-by-frame optimization process naturally encourage primitives to develop frame-specific preferences. Such frame preference can lead to noticeable fluctuations in test performance during late-stage training, as observed in Figure 4. Motivated by ensemble learning and SWA (Izmailov et al., 2018), we average the learned attribute parameters in the later stages of training. Unlike standard DNNs, where averaging is typically performed per epoch, Dynamic GS requires an iteration-based schedule. Specifically, we save an attribute snapshot every 500 iterations and average the last five snapshots, denoted by $\{\boldsymbol{\theta}^i\}$. We term this strategy Stochastic Attribute Averaging (SAA):

$$\boldsymbol{\theta}_{SAA} = \frac{\sum_i^n \boldsymbol{\theta}_i}{n} \quad (7)$$

## 5. Experiments

**Baselines and Configuration** We validate our method on two representative explicit Dynamic GS baselines with fundamentally different pipeline designs: 4DGS (Yang et al., 2024) and Ex4DGS (Lee et al., 2024). For each baseline, we preprocess the datasets using the official preprocessing scripts released with the corresponding project, ensuring consistent data preprocessing and initialization across all comparisons. Moreover, we use the original pipeline configurations and hyperparameters released by each method without modification. For 4DGS, the default $\epsilon$ is set to $10^{-15}$, inherited from the vanilla 3DGS implementation[1]; in contrast, Ex4DGS replaces Adam with RAdam and uses $\epsilon = 10^{-8}$, a commonly adopted choice in DNN training.

---

[1] https://github.com/graphdeco-inria/gaussian-splatting.git

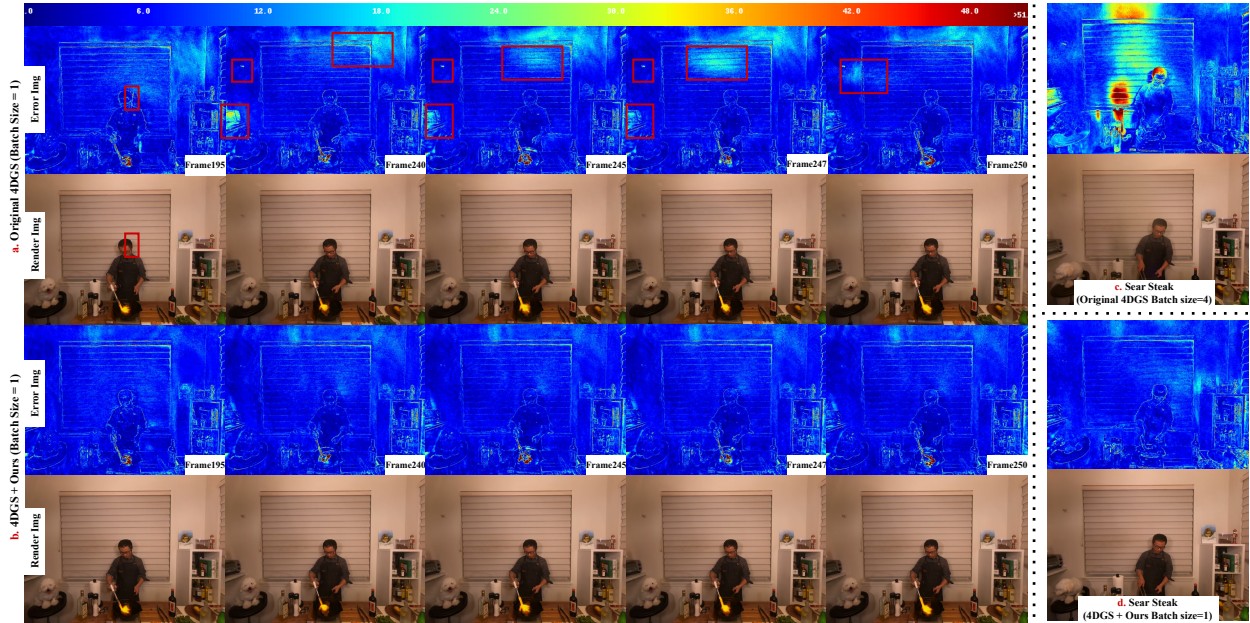

*Figure 3.* Qualitative comparisons of 4DGS with rendered images and error maps. The color-error mapping is shown on the top, with blue indicating lower error and red indicating higher error. a. Original 4DGS on Flame Steak from N3DV with batch size $\mathcal{B} = 1$; b. 4DGS with our method on Flame Steak with $\mathcal{B} = 1$; c. Original 4DGS on Sear Steak from N3DV with $\mathcal{B} = 4$; d. 4DGS with our method on Sear Steak with $\mathcal{B} = 1$.

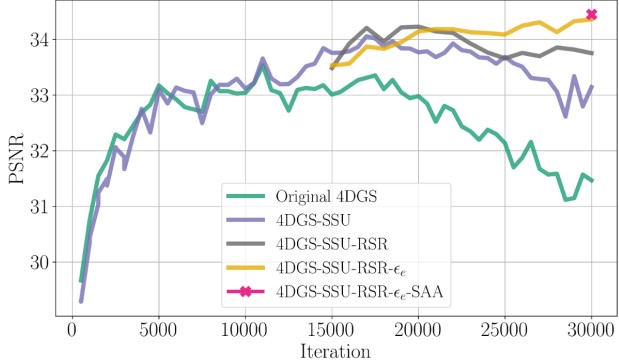

*Figure 4.* Comparison of test-set PSNR curves for different variants during 4DGS training.

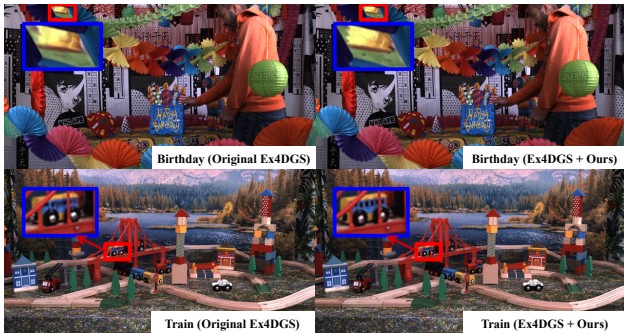

*Figure 5.* Qualitative comparison between the original Ex4DGS and Ex4DGS enhanced with our proposed strategy.

All experiments are conducted on a single NVIDIA RTX A6000 GPU with 48 GB of memory.

**Dataset and Metric** Following the official experimental protocols of both baselines, we evaluate our method on two *long-sequence real-world* datasets: Neural 3D Video (N3DV) (Li et al., 2022) and the Technicolor dataset (Sabater et al., 2017). For 4DGS, we report the main results on N3DV and conduct ablation studies on Flames Salmon and Cut Roasted Beef. For Ex4DGS, we evaluate on both datasets and conduct ablation studies on N3DV. We report PSNR, SSIM, MS-SSIM, and LPIPS on the held-out test views at the *last training iteration*. To better illustrate

the visual quality, we further provide error map visualizations. Each error map is obtained by directly computing the Euclidean distance in RGB space between the rendered image and the ground-truth image. The resulting error values are then mapped to the jet colormap, where blue regions indicate smaller errors and red regions indicate larger errors. The mapping between error values and colors is provided in Figure 3.

N3DV provides six indoor multi-view video sequences, each captured by 18 to 21 cameras at a resolution of $2704 \times 2028$ for 300 frames, corresponding to 10 seconds of video. Following the conventional evaluation protocol, both training and evaluation are conducted at half of the original resolu-

*Table 1.* Ablation study of different components of our method on 4DGS using N3DV scenes.

| Batch Size | SSU | RSR | $\epsilon$ | SAA | Cut Roasted Beef | | | | Flame Salmon | | | |
|---|---|---|---|---|---|---|---|---|---|---|---|---|
| | | | | | PSNR | SSIM | MS-SSIM | LPIPS(vgg) | PSNR | SSIM | MS-SSIM | LPIPS(vgg) |
| 4 | x | x | $10^{-15}$ | x | 32.239 | 0.952 | 0.977 | 0.139 | 28.992 | 0.922 | 0.957 | 0.142 |
| 4 | ✓ | x | $10^{-15}$ | x | 33.141 | 0.956 | 0.979 | 0.136 | 29.786 | 0.927 | 0.962 | 0.140 |
| 4 | ✓ | ✓ | $10^{-15}$ | x | 33.820 | 0.958 | 0.979 | 0.136 | 29.715 | 0.927 | 0.962 | 0.142 |
| 4 | ✓ | ✓ | $\epsilon_e$ | x | 34.202 | 0.958 | 0.980 | 0.136 | 29.794 | 0.926 | 0.961 | 0.141 |
| 4 | ✓ | ✓ | $\epsilon_e$ | ✓ | 34.321 | 0.960 | 0.981 | 0.135 | 29.891 | 0.928 | 0.962 | 0.140 |
| 1 | ✓ | ✓ | $\epsilon_e$ | ✓ | 33.617 | 0.958 | 0.979 | 0.145 | 29.419 | 0.925 | 0.961 | 0.150 |

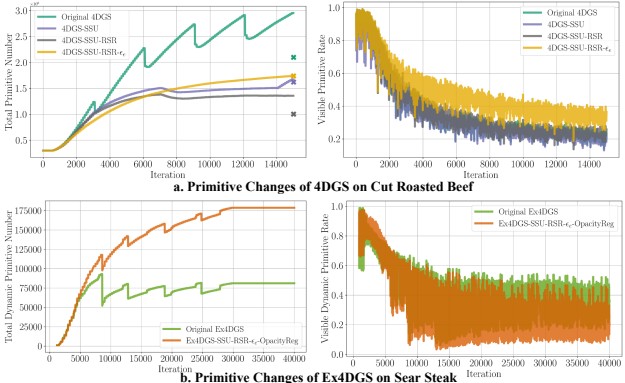

a. Primitive Changes of 4DGS on Cut Roasted Beef

b. Primitive Changes of Ex4DGS on Sear Steak

*Figure 6.* Evolution of the total number of primitives and the visible primitive rate, i.e., the fraction of visible primitives, for different variants during training. The symbol "×" in (a) denotes the final number of active primitives, where active primitives are defined as those with opacity greater than 0.05.

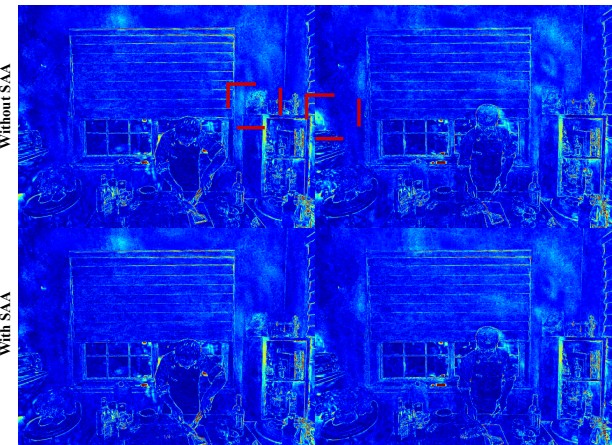

*Figure 7.* Error map visualizations of 4DGS with and without SAA.

tion, and the center camera is held out as a novel view for evaluation.

The Technicolor light field dataset consists of video sequences captured using a synchronized 4×4 camera array at a spatial resolution of 2048×1088. Evaluation is conducted on four distinct scenes, namely Birthday, Fabien, Theater, and Trains, at the original full resolution. In the official Ex4DGS configuration (Lee et al., 2024), only 50 frames per scene are used for evaluation; however, this short temporal span exhibits limited motion and does not fully reflect the long-sequence setting considered in our work. Therefore, we keep the same initial frame selection but extend the evaluation to 150 frames for all scenes, except Fabien, where we use 140 frames due to the limited sequence length.

## 5.1. Analysis

**Discussion on Regularization** We find that, although existing Dynamic GS methods can faithfully reconstruct the overall scene structure, they often suffer from generalization issues that manifest as localized blurring. We provide qualitative results for 4DGS and Ex4DGS in Figure 3 (c) and Figure 5, respectively. We first evaluate the effect of regularization on 4DGS. As shown in Figure 4, incorporating

SSU and regularization improves novel-view reconstruction quality; for Cut Roasted Beef, PSNR increases from 32.239 to 33.820. However, applying the same strategy to Ex4DGS while using the same $\epsilon$ as in 4DGS does not yield clear gains and can even cause substantial degradation on certain scenes, e.g., PSNR on Flame Steak from N3DV drops to 30.57. A similar limitation is also observed when applying the same strategy to 4DGS on more scenes. As analyzed in Section 4, we attribute this performance sensitivity primarily to the non-uniform scaling and characteristics of primitive gradients.

**Role of $\epsilon$.** To validate the effect of non-uniform scaling, we first study how different $\epsilon$ values affect the reconstruction quality of Ex4DGS. As analyzed in Section 4.2, the second-moment estimate in explicit Dynamic GS quickly stabilizes to a characteristic order of magnitude. Since Ex4DGS introduces two types of Gaussian primitives, we visualize the evolution of their second-moment estimates in Figure 13, which initially lie around $10^{-6}$ and $10^{-7}$, respectively. Guided by this observation, we evaluate $\epsilon \in \{10^{-6}, 10^{-7}\}$ and summarize the results in Table 2. We find that simply adjusting $\epsilon$ can improve reconstruction quality. Consistent with Section 4.2, increasing $\epsilon$ effectively induces a less adaptive update regime, making more primitives operate in this

*Table 2.* Ablation study of our proposed components integrated into Ex4DGS on N3DV.

| SSU | $\epsilon$ | RSR | Opacity Regularization | SAA | PSNR | SSIM | MS-SSIM | LPIPS(vgg) |
|---|---|---|---|---|---|---|---|---|
| x | $10^{-8}$ | x | x | x | 31.315 | 0.946 | 0.968 | 0.132 |
| ✓ | $10^{-15}$ | ✓ | ✓ | x | 31.267 | 0.948 | 0.969 | 0.135 |
| ✓ | $10^{-6}$ | x | x | x | 31.618 | 0.949 | 0.967 | 0.132 |
| ✓ | $10^{-7}$ | x | x | x | 31.458 | 0.949 | 0.969 | 0.129 |
| ✓ | $\epsilon_e$ | x | x | x | 31.729 | 0.951 | 0.970 | 0.128 |
| ✓ | $\epsilon_e$ | ✓ | x | x | 32.112 | 0.951 | 0.971 | 0.128 |
| ✓ | $\epsilon_e$ | ✓ | ✓ | x | 32.159 | 0.951 | 0.971 | 0.128 |
| ✓ | $\epsilon_e$ | ✓ | ✓ | ✓ | 32.204 | 0.951 | 0.971 | 0.127 |

*Table 3.* Quantitative results of Ex4DGS on the Technicolor dataset.

| | PSNR | SSIM | MS-SSIM | LPIPS(Alex) |
|---|---|---|---|---|
| Ex4DGS | 31.809 | 0.901 | 0.963 | 0.108 |
| +Ours(w/o SAA) | 32.179 | 0.906 | 0.964 | 0.101 |
| +Ours(w/ SAA) | 32.281 | 0.907 | 0.965 | 0.101 |

*Table 4.* Quantitative results for 4DGS on N3DV. [†]: For Flame Steak and Sear Steak, we evaluate our method only with batch size $\mathcal{B} = 1$, while the other four scenes use batch size $\mathcal{B} = 4$.

| | Batch size | PSNR | SSIM | MS-SSIM | LPIPS(vgg) |
|---|---|---|---|---|---|
| 4DGS | 4 | 30.202 | 0.943 | 0.968 | 0.140 |
| +Ours[†] | - | 32.469 | 0.949 | 0.974 | 0.138 |
| +Ours | 1 | 32.174 | 0.947 | 0.973 | 0.145 |

regime. Such constrained adaptivity is particularly beneficial for scenes with larger motion. For example, with $\epsilon = 10^{-6}$, PSNR improves from 32.332 to 32.734 on Cook Spinach and from 33.037 to 33.660 on Sear Steak, as shown in Appendix Section F.

**Effectiveness of Non-Adaptive Step**   Although adjusting $\epsilon$ can control the transition toward a less adaptive update regime and improve reconstruction quality, the effective adaptive step size of each primitive is also influenced by multiple factors, such as primitive design, image scale, and dataset characteristics. Consequently, the optimal $\epsilon$ is method- and scene-dependent. Motivated by the characteristics of primitive gradients, we propose a fast $\epsilon$ estimation method that assigns a type-specific $\epsilon_e$ to different primitive types. Quantitative results are summarized in Table 2. With a better-calibrated $\epsilon$, reconstruction quality is further improved. Moreover, with a suitable $\epsilon_e$, regularization also becomes effective, improving PSNR on N3DV from 31.315 to 32.159. We further evaluate Ex4DGS on the Technicolor dataset, where our method improves PSNR from 31.809 to 32.179, as shown in Table 3. The estimated $\epsilon_e$ also yields consistent gains when applied to 4DGS on N3DV. For Cut Roasted Beef, the estimated $\epsilon_e$ brings an additional PSNR gain of 0.4.

To further understand the role of non-adaptive updates, we analyze the distribution of primitives under different methods. In Dynamic GS, reconstructing moving objects naturally yields primitives that exist only over limited temporal intervals rather than throughout the entire video; according to the estimated temporal probability, their opacity can be effectively scaled to near zero. During training, we measure the *Visible Primitive Rate*, defined as the ratio of primitives

that are active and visible from the current viewpoint at the current time step to the total number of primitives. This metric serves as a proxy for the temporal distribution of primitives. We find that, unlike regularization alone, introducing non-adaptive updates can substantially alter this distribution. As shown in Figure 6, for 4DGS, which typically reconstructs scenes with more primitives, non-adaptive updates enable reconstruction with fewer primitives while maintaining their effectiveness over longer temporal spans. In contrast, using RSR alone neither changes the Visible Primitive Rate nor preserves effective primitives, and it significantly reduces the number of active primitives during training. Notably, the effect of non-adaptive updates is not uniform across pipelines: for Ex4DGS on N3DV, it encourages the use of more primitives for representation while reducing their visible rate.

**Frame Preference and SAA.**   We argue that frame-by-frame optimization in Dynamic GS tends to induce frame-preferential representations. As shown in Figure 4, this effect is reflected by noticeable test-set fluctuations across iterations, indicating iteration-dependent frame preferences. Batch sampling partially mitigates this frame preference by averaging gradients across frames. To further counter this inherent bias, we introduce SAA, which is conceptually analogous to batch sampling but averages attribute parameters across iterations instead. For 4DGS with batch sampling, SAA still yields a PSNR gain of 0.1. For 4DGS variants without batch sampling, e.g., Sear Steak in Appendix Table 16, the improvement can reach 0.2. For Ex4DGS on N3DV, SAA does not consistently improve all scenes but does not degrade reconstruction quality; on Technicolor, SAA further improves reconstruction performance. We also

provide qualitative comparisons in Figure 7, showing that the errors in certain regions are substantially reduced after applying SAA.

**4DGS without Batch Sampling.** Although we validate the effectiveness of our method on several N3DV scenes, including Flame Salmon, Cut Roasted Beef, Coffee Martini, and Cook Spinach, we observe that PSNR can still drop substantially on scenes such as Flame Steak and Sear Steak when batch sampling is enabled. In 4DGS, the temporal distribution is typically modeled as a Gaussian distribution. With batch sampling, gradient averaging reduces the estimated second moment, particularly suppressing extreme values. Under this suppression, 4DGS tends to produce widespread blurring at the end of training, as shown in Figure 3(c). In contrast, disabling batch sampling introduces frame-wise artifacts, as shown in Figure 3(a). Since our method alters the primitive distribution, we further evaluate 4DGS with the batch size set to 1. We find that our method effectively suppresses these artifacts without relying on batch sampling, as shown in Figure 3(b). Moreover, removing batch sampling reduces the training time by approximately 50% under the same framework.

## 6. Conclusion and Discussion

This work targets generalization degradation in explicit Dynamic GS. Guided by insights from DNN optimization and the optimization characteristics of Gaussian primitives, we revisit two factors: non-uniform update scaling induced by adaptive optimizers and motion-induced frame preference in the frame-by-frame optimization process. To address these factors, we propose an update-mixture strategy consisting of Strictly Sparse Update, Mix-Adaptive Update Step, and Stochastic Attribute Averaging. Extensive experiments show that introducing non-adaptive updates can reshape the primitive distribution and improve dynamic-scene modeling. By averaging attribute parameters across iterations to alleviate frame preference, SAA further brings consistent gains in reconstruction quality and helps recover finer image details.

**Limitation.** Although our method improves the optimization of explicit Dynamic GS, its effectiveness can be limited when the baseline is trained with an unsuitable learning rate. A well-calibrated learning rate not only improves the reconstruction quality of the baseline, but is also critical for our method to take effect. This observation also highlights the important role of the update step in Dynamic GS optimization, which we further discuss in Sec. D.3. For operations that may inherently introduce generalization issues, such as batch sampling, our method is not always effective. Nevertheless, it still brings significant improvements even without batch sampling. Finally, due to the lack of strong geometric priors, our method remains challenged by extremely large

motions. Failure cases are provided in Figure 11 in the appendix.

## Acknowledgements

This work was supported by China Mobile Hunan Company Limited, China Mobile Communications Group Co., Ltd., and by the National Natural Science Foundation of China under Grant U22B2050, 62425305, 62221002, and 62503161. This work was conducted as part of the project "Networked Robotic System for Major Equipment Manufacturing (5G+Robotics)".

## Impact Statement

This paper presents work whose goal is to advance the field of Machine Learning. There are many potential societal consequences of our work, none which we feel must be specifically highlighted here.

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

## A. Additional Evidence for Generalization Issues

**Testset Evaluation in 4DGS**    We find that, in the current 4DGS implementation, the model is *evaluated on the test set every 500 iterations*, and the *attribute set with the best test-set performance* is retained. We consider this design unreasonable for two reasons. First, it violates standard machine learning practice, where the test set should not be used for model selection. Second, test-set evaluation is extremely expensive in Dynamic Gaussian Splatting, since the test set must cover an entire temporal sequence. In our practical runs, a single evaluation can take more than 30 minutes. We also experimented with sparser evaluation intervals; however, under sparse evaluation, the retained attribute set often exhibits relatively poor performance.

Overall, we argue that this implementation choice is inappropriate. A more appropriate practice is to mitigate the overfitting, or generalization, issue so that, as in vanilla 3DGS, the final iteration can still yield a high-quality attribute set.

The corresponding implementation can be found in lines 219–223 of the `train.py` file in the official repository[2]. In fact, a closely related concern has also been raised by the community[3]. In contrast, that work reports the attributes from the final iteration, as shown by Ours(30k) and Ours(full) in Table 1 of the original paper (Liu et al., 2025).

```
1    ...
2    test_psnr = training_report(...)
3    if (iteration in testing_iterations):
4    if test_psnr >= best_psnr:
5    best_psnr = test_psnr
6    print("\n[ITER {}] Saving best checkpoint".format(iteration))
7    torch.save((gaussians.capture(), iteration), scene.model_path + "/chkpnt_best.pth")
8    ...
```

**Same Issue Reported in Ex4DGS Project Page**    We also observe that, on the Ex4DGS GitHub issue page[4], several users have reported that directly running the released code does not reproduce the performance reported in the paper. Moreover, according to the implementation, which performs validation at 30,000 and 40,000 iterations, the PSNR obtained at the final iteration, i.e., 40,000 iterations, can be significantly lower than that obtained at 30,000 iterations. This indicates that the generalization issue discussed in our paper is not an isolated or accidental phenomenon. Related reports can be found mainly in Issues #3 and #10.

In light of this finding, all of our implementations are reproduced locally, and we ensure that the same initial point cloud is used throughout.

## B. Preliminary Detail

Gaussian primitives are employed to model static scenes and are parameterized by a set of attributes, including position $\boldsymbol{\mu}$, covariance $\boldsymbol{\Sigma}$, opacity $o$, and color $\boldsymbol{c}$. Given a viewpoint $v$ and the associated camera transformation, each primitive is transformed into the camera coordinate system and projected onto the image plane, producing a 2D Gaussian footprint $\mathcal{G}_i(\boldsymbol{p}; \boldsymbol{\mu}_i^v, \boldsymbol{\Sigma}_i^v)$ at pixel $\boldsymbol{p}$. The pixel color is then rendered via $\alpha$-blending:

$$C(\boldsymbol{p}) = \sum\nolimits_{i=1}^{N_p^v} \boldsymbol{c}_i^v o_i \mathcal{G}_i(\boldsymbol{p}; \boldsymbol{\mu}_i^v, \boldsymbol{\Sigma}_i^v) T_i,$$
$$\text{where} \quad T_i = \prod\nolimits_j^{i-1} (1 - o_j \mathcal{G}_j(\boldsymbol{p}; \boldsymbol{\mu}_j^v, \boldsymbol{\Sigma}_j^v)) \tag{8}$$

Here, $N_p^v$ denotes the number of Gaussian primitives that contribute to pixel $p$ under viewpoint $v$. The indices $i$ and $j$ correspond to primitives sorted by rendering depth. $\boldsymbol{c}^v$, $\boldsymbol{\mu}^v$, and $\boldsymbol{\Sigma}^v$ denote the corresponding attributes after applying the camera transformation (Kerbl et al., 2023).

To model dynamic scenes, a straightforward way in explicit Dynamic GS is to introduce an additional temporal dimension

---

[2]https://github.com/fudan-zvg/4d-gaussian-splatting
[3]https://github.com/RongLiu-Leo/beta-splatting/issues/13
[4]https://github.com/juno181/Ex4DGS/issues

and generalize Eq. 8 to Eq. 9.

$$C(\boldsymbol{p}, t) = \sum_{i=1}^{N_p^v} \boldsymbol{c}_i^v o_i(t) \mathcal{G}_i(\boldsymbol{p}; \boldsymbol{\mu}_i^v(t), \boldsymbol{\Sigma}_i^v(t)) T_i \tag{9}$$

where $o_i(t)$ is related to the appearance or disappearance of a primitive and is commonly instantiated as an opacity attribute modulated by a time-dependent probability (Yang et al., 2024; Lee et al., 2024). Meanwhile, $\boldsymbol{\mu}_i(t)$ and $\boldsymbol{\Sigma}_i(t)$ govern the temporal evolution of geometry and anisotropy, respectively. Notably, 4DGS and Ex4DGS instantiate these time-varying parameterizations in fundamentally different ways.

### B.1. 4DGS

4DGS (Yang et al., 2024) follows a pipeline largely similar to that of vanilla 3DGS (Kerbl et al., 2023), while extending each 3D Gaussian primitive to a 4D Gaussian primitive. Accordingly, the covariance matrix $\boldsymbol{\Sigma} \in \mathbb{S}_+^{4 \times 4}$ is parameterized by a scaling matrix $\boldsymbol{S} = \mathrm{diag}(s_1, s_2, s_3, s_4), s_1, s_2, s_3, s_4 > 0$, and a 4D rotation matrix $\boldsymbol{R} \in \{\mathbb{R}^{4 \times 4} | \boldsymbol{R}^\top \boldsymbol{R} = \boldsymbol{I}, \det(\boldsymbol{R}) = 1\}$. Leveraging standard properties of multivariate Gaussians, we can then derive the conditional 3D Gaussian as:

$$\boldsymbol{\Sigma} = \boldsymbol{R}\boldsymbol{S}\boldsymbol{S}^\top \boldsymbol{R}^\top \tag{10}$$

$$\boldsymbol{\mu}(t) = \boldsymbol{\mu} + \boldsymbol{\Sigma}_{1:3,4}\boldsymbol{\Sigma}_{4,4}^{-1}(t - \mu_t) \tag{11}$$

$$\boldsymbol{\Sigma}' = \boldsymbol{\Sigma}_{1:3,1:3} - \boldsymbol{\Sigma}_{1:3,4}\boldsymbol{\Sigma}_{4,4}^{-1}\boldsymbol{\Sigma}_{4,1:3} \tag{12}$$

$$o(t) = o\mathcal{N}(t; \mu_t, \boldsymbol{\Sigma}_{4,4}) \tag{13}$$

where $\mathcal{N}(\cdot)$ denotes Gaussian function.

Given the camera transformation associated with viewpoint $v$, the color of pixel $\boldsymbol{p}$ on the screen is rendered via $\alpha$-blending:

$$C(\boldsymbol{p}, t) = \sum_{i=1}^{N_p^v} \boldsymbol{c}_i^v(t) o_i(t) \mathcal{G}_i(\boldsymbol{p}; \boldsymbol{\mu}_i^v(t), \boldsymbol{\Sigma}_i'^v) T_i \tag{14}$$

Here, $N_p^v$ denotes the number of Gaussian primitives that contribute to pixel $p$ under viewpoint $v$. The indices $i$ and $j$ denote primitives sorted by rendering depth. All attributes are transformed by the corresponding viewpoint-dependent transformation before rendering. 4DGS also employs 4D spherindrical harmonics to extend the color representation into a time-conditioned form, $\boldsymbol{c}(t)$.

4DGS adopts the same densification procedure as vanilla 3DGS (Kerbl et al., 2023), which includes cloning, splitting, and pruning. In particular, pruning removes primitives whose opacity falls below a predefined threshold, as these primitives are considered redundant. To reliably identify redundant primitives, 4DGS also inherits the opacity reset mechanism, which periodically multiplies primitive opacity by a very small factor, typically driving it close to zero.

To reduce temporal flickering and jitter and obtain more reliable gradient estimates, 4DGS also employs **batch sampling**. Specifically, given a batch size $\mathcal{B}$, it updates Gaussian attributes using gradients averaged over the sampled frames.

$$\boldsymbol{g}_k = \frac{1}{\mathcal{B}} \sum_{\mathcal{B}} \nabla \ell \tag{15}$$

where $\ell$ denotes any loss function.

### B.2. Ex4DGS

Ex4DGS (Lee et al., 2024) introduces improvements over vanilla 3DGS in terms of primitives, the pipeline, and the optimization.

In terms of primitive design, Ex4DGS first partitions Gaussian primitives into **static primitives** and **dynamic primitives**. Static primitives follow the formulation of vanilla 3DGS, except that their position is reformulated as:

$$\boldsymbol{\mu}(t) = \boldsymbol{\mu} + (\frac{t}{l} - \mu_t)\boldsymbol{d} \tag{16}$$

where $l$ is the duration of a scene and $\boldsymbol{d}$ is a trainable vector parameter.

Dynamic primitives are introduced to represent complex object motion, and Ex4DGS models the temporal evolution of both position and rotation via interpolation. Specifically, Ex4DGS defines a set of keyframes $\mathcal{K} = \{t \mid t = nI, \ n \in \mathbb{Z}, \ t \in \Gamma\}$, where $I$ denotes the keyframe interval and $\Gamma$ is the set of timestamps. Both position and rotation are then extended to sequences $\{\boldsymbol{x}_i\}_0^n$ and $\{\boldsymbol{r}_i\}_0^n$, respectively, and expressed through interpolation between these keyframes:

$$\boldsymbol{\mu}(t) = \mathrm{CHip}(\boldsymbol{x}_n, \frac{\boldsymbol{x}_{n+1} - \boldsymbol{x}_{n-1}}{2I}, \boldsymbol{x}_{n+1}, \frac{\boldsymbol{x}_{n+2} - \boldsymbol{x}_n}{2I}; \frac{t - nI}{I}) \quad \text{and} \quad n = \lfloor \frac{t}{I} \rfloor \tag{17}$$

$$\boldsymbol{q}(t) = \mathrm{Slerp}(\boldsymbol{r}_n, \boldsymbol{r}_{n+1}; \frac{t - nI}{I}) \quad \text{and} \quad n = \lfloor \frac{t}{I} \rfloor \tag{18}$$

where $\mathrm{CHip}(\cdot, \cdot, \cdot, \cdot; \cdot)$ and $\mathrm{Slerp}(\cdot, \cdot; \cdot)$ are Cubic Hermite Interpolator and Spherical Linear Interpolation respectively. $\boldsymbol{x}_i$ is the primitive position at keyframe $i$; $\boldsymbol{r}_n$ is the unit quaternion at keyframe $i$ and the $\boldsymbol{q}(t)$ can be restored to rotation matrix. *The interpolation-based design implies that, although complex motion of a primitives is represented by a sequence with $n$ elements*, **at any timestamp $t$ the pipeline accesses only a small subset of elements from this sequence**.

Ex4DGS also uses a probability function $\sigma(t)$ to model the appearance and disappearance of primitives, and defines the time-dependent opacity as $o(t) = o\sigma(t)$. Finally, given the camera transformation associated with viewpoint $v$, the color of screen pixel $\boldsymbol{p}$ is rendered via $\alpha$-blending:

$$C(\boldsymbol{p}, t) = \sum_{i=1}^{N_p^v} \boldsymbol{c}_i^v o_i(t) \mathcal{G}_i(\boldsymbol{p}; \boldsymbol{\mu}_i^v(t), \boldsymbol{\Sigma}_i^v(t)) T_i \tag{19}$$

Here, $N_p^v$ denotes the number of Gaussian primitives that contribute to pixel $p$ under viewpoint $v$. The indices $i$ and $j$ index primitives sorted by rendering depth. All attributes are preprocessed by the corresponding transformation before rendering. For static primitives, $o(t)$ and $\boldsymbol{\Sigma}(t)$ are time-invariant and can be expressed as constant functions.

Ex4DGS also introduces a dedicated densification stage during training, including cloning, splitting, and pruning. Since filtering unnecessary dynamic primitives is non-trivial in a temporal context, Ex4DGS additionally tracks reconstruction errors and prunes primitives according to accumulated reconstruction errors. Accordingly, Ex4DGS does not employ opacity reset in its pipeline. Moreover, to accommodate its primitive design, Ex4DGS incorporates an additional dynamic-primitive extraction step that derives dynamic primitives from static ones. To stabilize training under fast object motion or abrupt disappearance, Ex4DGS further adopts a **progressive training scheme**: it first learns from a short initial segment of the input video and then gradually extends the temporal span. Consequently, in the early stages of training, the model represents the scene only over a limited time horizon.

Ex4DGS adopts the same objective functions as vanilla 3DGS, but uses RAdam (Liu et al., 2020) as the optimizer. It further introduces an additional **motion regularization** term on primitive trajectories to improve reconstruction quality. Concretely, this regularization penalizes the motion magnitude $\|\boldsymbol{d}\|$ as well as inter-frame displacements $\|\boldsymbol{x}_{i+1} - \boldsymbol{x}_i\|$ over the entire trajectory $\{\boldsymbol{x}_i\}$. These regularization terms are weighted with coefficients on the order of $10^{-4}$.

# C. More Analysis

## C.1. Motion of the Scenes

We visualize several representative frames from the Cut Roasted Beef and Flame Salmon scenes in N3DV (Li et al., 2022), together with their corresponding optical flow estimates obtained using RAFT (Teed & Deng, 2020). We observe that Cut Roasted Beef exhibits significantly larger motion, with substantial motion distributed across the entire target region. In contrast, although motion is also present in Flame Salmon, it is mainly confined to one hand of the subject and occurs within a limited spatial and temporal range. Accordingly, in this paper, we categorize Cut Roasted Beef as a large-motion scene and Flame Salmon as a small-motion scene.

For the N3DV dataset, we further observe that Coffee Martini, similar to Flame Salmon, can also be categorized as a small-motion scene, whereas the remaining four scenes—Cook Spinach, Cut Roasted Beef, Flame Steak, and Sear Steak—are characterized by large target motion. It is worth noting that our discussion is limited to the frames used for training. For example, in the Flame Salmon scene, only the first 300 frames are used for training; although larger motions appear in later frames, they are beyond the scope of our analysis, as we strictly follow the training split of the baseline.

## C.2. Unstability in Opacity Reset

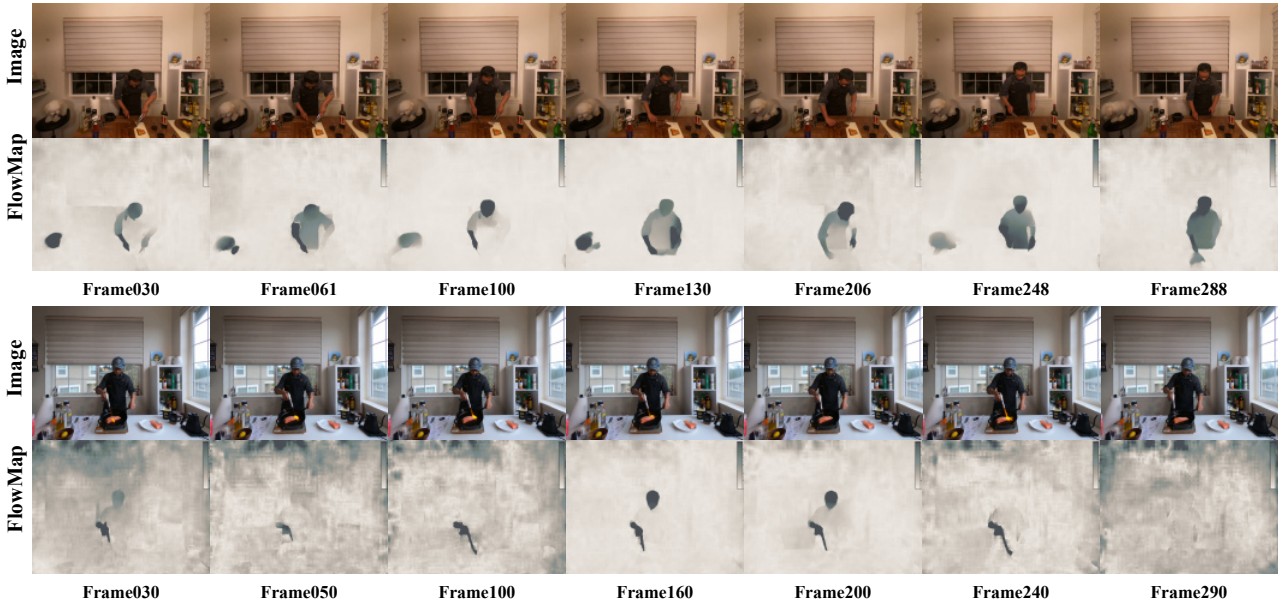

*Figure 8.* Visualization of representative scene frames and their corresponding optical flow estimates.

When opacity reset is applied, the opacity of all primitives is scaled to a very small value (Kerbl et al., 2023), with the intent of facilitating the pruning of redundant primitives. However, recent studies indicate that this operation can destabilize optimization (Rota Bulò et al., 2024). We further provide a visual example in Fig. 9, which illustrates the abrupt, large-magnitude change induced by opacity reset.

### C.3. Motion Regularization in Ex4DGS

We focus on the motion regularization for dynamic primitives, as the analysis for static primitives follows analogously. Using the notation introduced in Section B.2, the motion regularization for dynamic primitives is given by:

$$\mathcal{R}_m = \frac{\lambda}{N_p}\|\boldsymbol{x}_{n+1} - \boldsymbol{x}_n\|_2 \tag{20}$$

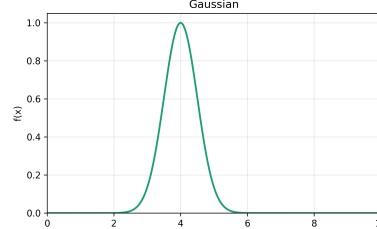

*Figure 9.* 1st moment in 4DGS with opacity reset (Iteration begins from 1000).

where $\lambda$ is typically set to $10^{-4}$ in practice, and $N_p$ denotes the average number of dynamic primitives. For N3DV, $1/N_p$ is usually on the order of $10^{-5}$. Consequently, the magnitude of $\nabla\mathcal{R}_m$ is on the order of $10^{-9}$, which is far smaller than the typical magnitude of $\sqrt{v}$. Moreover, since the gradient scale is affected by $T$, $\mathcal{R}_m$ may improperly penalize background primitives. Prior work interprets 3DGS as fitting a Gaussian Mixture Model (Charatan et al., 2024), a highly non-convex problem typically solved via Expectation–Maximization (Jin et al., 2016); therefore, suboptimal reconstruction induced by positional perturbations may be difficult to recover.

### C.4. Batch Sampling

Since the gradient scale of Gaussian primitives is coupled to opacity, and opacity in Dynamic GS is further modulated by an additional sparsity-inducing probability term, dynamic primitives typically contribute gradients only over a limited temporal range, as illustrated by the probability profile in Figure 10. Under batch sampling, averaging gradients across timestamps suppresses gradient magnitudes, which in turn reduces the estimated second moment. We argue that this effect can further exacerbate the generalization issue. We further sample the

*Figure 10.* Primitives that model dynamic objects typically exist only over a limited portion of the temporal axis; an example temporal probability profile of a primitive is illustrated here. Additional examples can be found in Figure 9 of (Yang et al., 2024).

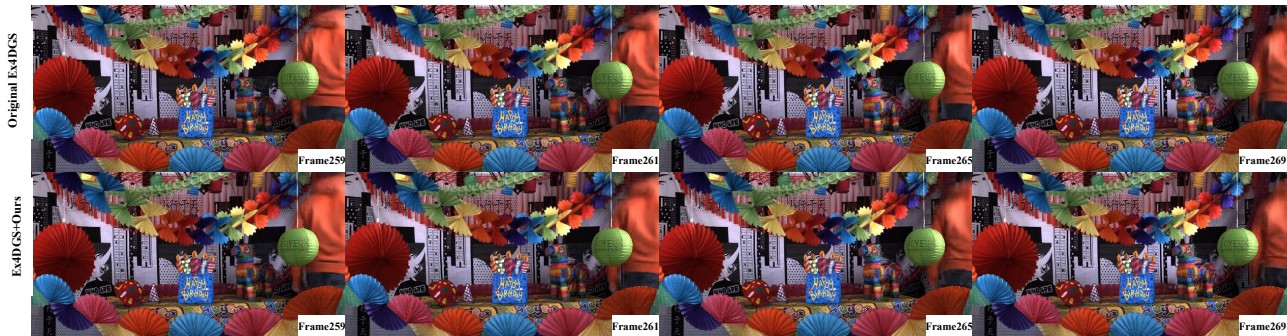

*Figure 11.* Visualization of rendering results in large motion.

optimizer states, and the corresponding information is shown in Fig. 11. When batch sampling is used, relatively large update steps may occur.

### C.5. Overhead Analysis of SAA

SAA introduces negligible computational overhead, since it only performs element-wise addition and averaging over parameters collected from several designated checkpoints. The averaged parameters are used only to obtain the final model and are not involved in subsequent optimization. Therefore, SAA does not require additional backward passes, gradient computation or rendering.

SAA can be implemented either offline or online. In the offline implementation, the selected checkpoints are saved during training and averaged after optimization. This strategy introduces no additional GPU memory consumption during training, apart from the disk storage needed for the saved checkpoints. Importantly, computing SAA from the stored checkpoints requires less memory than the original training process. Dynamic GS methods are typically optimized with adaptive-gradient optimizers, which maintain first- and second-moment states for each optimizable parameter. Offline SAA, however, only needs to load and average the checkpointed parameters, without storing gradients or optimizer states. As a result, its memory footprint is lower than the optimizer-state memory footprint required during training.

Alternatively, SAA can be implemented online by maintaining a running average on the GPU. This introduces additional persistent memory proportional to the size of the Gaussian parameters. However, after densification terminates, the number of primitives usually remains fixed unless extra pruning is applied, so the memory overhead becomes bounded and fixed. In our measurement on the Cut Roasted Beef scene with 2.99 million primitives and a batch size of 4, online SAA increases persistent GPU memory by about 1.8 GB, while the averaging step reaches a peak additional allocation of about 4.8 GB, measured by `torch.cuda.memory_allocated()` and `torch.cuda.max_memory_allocated()`, respectively.

## D. Implementation Detail

### D.1. Time Condition $\mathcal{T}(t)$ for Ex4DGS

Although the probability function $\sigma(t)$ in Ex4DGS can, in principle, indicate whether a primitive is disappearing, evaluating $\sigma(t)$ introduces additional computational overhead due to its piecewise form and the Gaussian terms involved in its definition. Following the efficient primitive handling strategy in (Papantonakis et al., 2024), we instead use $o(t)$ as a fast proxy and classify primitives with $o(t) < 0.05$ as disappearing primitives.

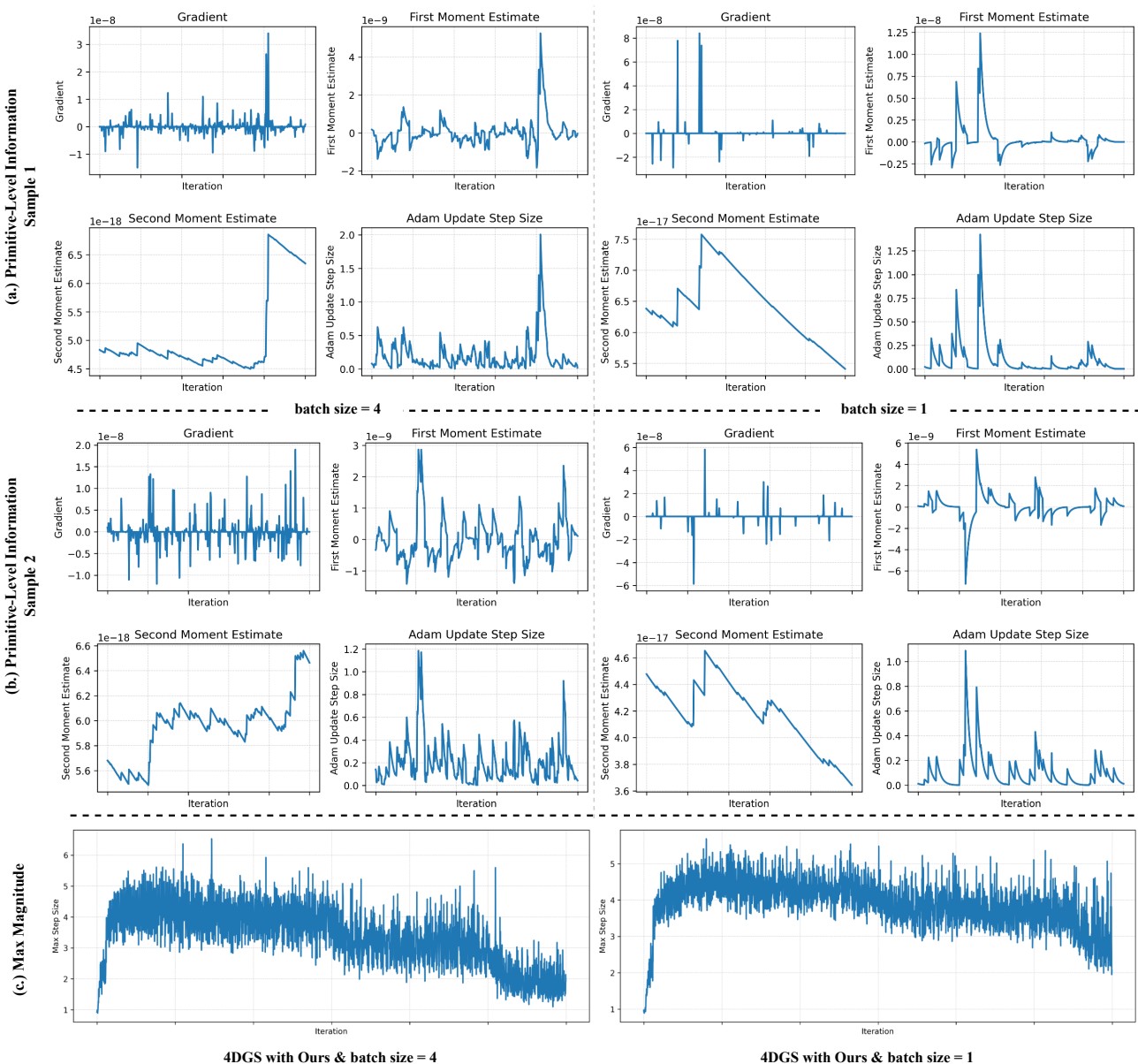

*Figure 12.* (a) and (b) show the optimizer-related quantities of two sampled primitives over a range of iterations after the densification stage. The batch-size-4 and batch-size-1 settings use the same set of primitives, where the batch-size-1 experiment is initialized from the primitives obtained under batch size 4 and warmed up for 500 iterations. (c) reports the evolution of the maximum value during training under batch size 4 and batch size 1.

## D.2. Point Clouds Initialization for N3DV in Ex4DGS

Following the discussion in Issue #3 of the Ex4DGS project[5], we replace the default point-cloud initialization for N3DV with the initialization tool provided by 4DGaussians (Wu et al., 2024).

## D.3. Learning Rate for Ex4DGS

We observe that Ex4DGS employs substantially different hyperparameter settings, including opacity-related learning rates and densification thresholds. For dynamic primitives, Ex4DGS mainly provides two configurations for the opacity,

---

[5]https://github.com/juno181/Ex4DGS/issues/3

opacity-center, and opacity-variance learning rates: $\{0.0001, 0.001, 0.001\}$ and $\{0.05, 0.001, 0.0001\}$. We find that the first configuration, which uses a smaller opacity learning rate, does not degrade reconstruction quality and can even improve it. Moreover, our method is effective only with this smaller-opacity-learning-rate configuration. Therefore, in all experiments on Technicolor, we use $\{0.0001, 0.001, 0.001\}$ for all scenes.

*Table 5.* Learning-rate ablation of Ex4DGS on the Fabien and Theater scenes from the Technicolor dataset.

| Learning Rate Configuration | Fabien | | | | Theater | | | |
|---|---|---|---|---|---|---|---|---|
| | PSNR | SSIM | MS-SSIM | LPIPS(vgg) | PSNR | SSIM | MS-SSIM | LPIPS(vgg) |
| Larger Learning Rate $\{0.05, 0.001, 0.0001\}$ | 34.475 | 0.881 | 0.952 | 0.319 | 30.459 | 0.868 | 0.943 | 0.273 |
| Smaller Learning Rate $\{0.0001, 0.001, 0.001\}$ | 34.735 | 0.882 | 0.953 | 0.320 | 30.400 | 0.867 | 0.943 | 0.275 |

One issue worth considering is that, although our method fails under the larger-opacity-learning-rate configuration, this actually reflects the influence of the learning step. As discussed in Section 4.2, Scale of Gradients (ii&iii), opacity affects both the scale and the spatial relationships of gradients. The two ablated opacity learning rates differ by $500\times$; since our method directly targets the adaptive update, such a large gap can naturally invalidate its effect. Our ablation shows that, under the smaller-opacity-learning-rate setting, reconstruction quality does not degrade, and our method still brings further improvements. We therefore believe that this comparison further highlights the role of the update step in Dynamic GS, which is consistent with our central discussion.

### D.4. $\epsilon$ Estimate in Dynamic Gaussian Splatting

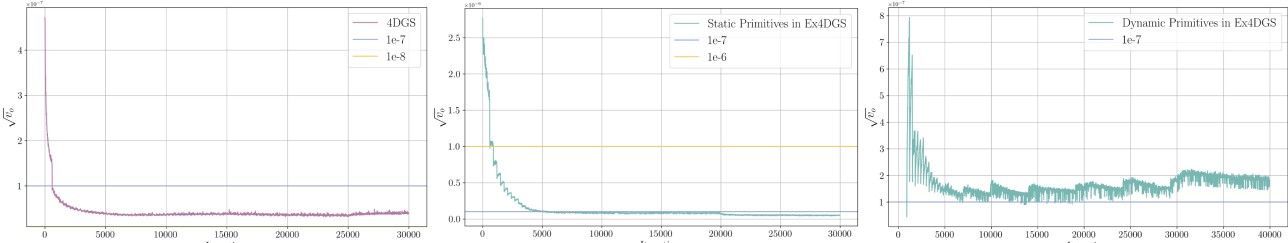

*Figure 13.* The 2nd moment for different primitives during training.

As shown in Fig. 13, the second-moment statistics quickly settle to an order of magnitude around $10^{-6}$ at the early stage. Accordingly, we set $\epsilon_e = 10^{-6}$ for static primitives and $\epsilon_e = 10^{-7}$ for dynamic primitives, since the opacity of dynamic primitives is additionally modulated by a temporal probability term. For scenes with small motion, we use smaller $\epsilon_e$ values for both static and dynamic primitives. For 4DGS, the corresponding second-moment statistics are initially around $10^{-7}$; since all primitives are probability-modulated, we set $\epsilon_e = 10^{-8}$.

### D.5. Regularization in this work

**ReState Regularization (RSR)**: Given the adaptive first moment $m_k$ and second moment $v_k$, we consider using RSR to attenuate the memory effect. Specifically, at a predefined iteration interval, we sample a subset of primitives and reset the corresponding $m_k$ and $v_k$ for the sampled primitives. Concretely, we obtain updated moments $m_k^{\text{new}}$ and $v_k^{\text{new}}$ via Eq. 21. $\alpha_1^2 = \alpha_2$ is enforced to preserve the magnitude of the update step. We adopt the default setting from the original paper, namely $\alpha_1 = 0.2$ and $\alpha_2 = 0.04$. Under this mechanism, the sampled primitives are continuously subjected to a memory penalty.

$$m_k^{\text{new}} = \alpha_1 m_k, \qquad v_k^{\text{new}} = \alpha_2 v_k \tag{21}$$

**Opacity regularization**: We employ a decoupled opacity regularization in our method. Specifically, on top of the original optimizer update, we compute the magnitude of the opacity regularization according to Eq. 22. The resulting term does not participate in the original gradient computation; instead, it is directly added to the update step produced by the optimizer,

after which the parameters are updated according to Eq. 23.

$$\nabla \mathcal{R}(o) = \min(\lambda_o \frac{\nabla o/N_I}{\sqrt{\hat{v}(o)'_t} + \epsilon}, \text{Const}) \tag{22}$$

$$\theta(o)_{k+1} = \theta(o)_k - \eta \times [\frac{\hat{m}(o)_k}{\sqrt{\hat{v}(o)_k} + \epsilon} + \nabla R(o)] \tag{23}$$

## E. Related work

### E.1. Opacity Optimization in 3DGS

Several studies have examined opacity optimization in 3DGS. (Fan et al., 2024) observe a large number of low-opacity primitives, indicating ineffective pruning. (Wang et al., 2025a) revisit primitive transportation in 3DGS and attribute local minima to vanishing opacity gradients, which prevent ineffective primitives from being pruned. (Rota Bulò et al., 2024) show that the original opacity reset in 3DGS is unstable and propose a constant opacity decay in Eq. 24, a strategy also adopted in (Han et al., 2024) to mitigate view inconsistency. (Han et al., 2024) state that 3DGS flexibly optimizes the scale and opacity during training, leading to artifacts in novel-view renderings. Building on this line of work, 3DGSMCMC (Kheradmand et al., 2024) introduces an $\ell_1$-norm opacity regularization (Eq. 25), which improves performance over constant decay and has been widely adopted (Liu et al., 2025; Zhu et al., 2025). Inspired by gradient decoupling in AdamW (Loshchilov & Hutter), (Ding et al., 2026) further proposes a decoupled opacity regularization in Eq. 22 for more effective penalization. (Chen et al., 2026) introduce opacity noise injection to alleviate overfitting in sparse-view 3DGS, which is not suggested by (Kheradmand et al., 2024). Overfitting is typically considered negligible in dense-view 3DGS. Overall, opacity decay and $\ell_1$-norm regularization remain the most widely adopted and generalization-oriented strategies, while more complex opacity losses rely on stronger assumptions and see limited use (Jiang et al., 2024).

$$o_{\text{new}} = \lambda_{\text{decay}} o, \quad 0 < \lambda_{\text{decay}} < 1 \tag{24}$$

$$\mathcal{R}_{\ell_1}(o) = \lambda_o |o| \tag{25}$$

## F. Results

### F.1. Flame Salmon from N3DV

Table 6. Ablation study of 4DGS on the Flame Salmon from the N3DV dataset.

| Batch Size | SSU | RSR | $\epsilon$ | SAA | PSNR | SSIM | MS-SSIM | LPIPS(vgg) |
|---|---|---|---|---|---|---|---|---|
| 4 | x | x | $10^{-15}$ | x | 28.992±0.058 | 0.922±0.001 | 0.957±0.000 | 0.142±0.000 |
| 4 | ✓ | ✓ | $\epsilon_e (10^{-8})$ | x | 29.794±0.033 | 0.926±0.001 | 0.961±0.000 | 0.141±0.001 |
| 4 | ✓ | ✓ | $\epsilon_e (10^{-8})$ | ✓ | 29.891±0.036 | 0.928±0.000 | 0.962±0.000 | 0.140±0.000 |
| 1 | ✓ | ✓ | $\epsilon_e (10^{-8})$ | ✓ | 29.419±0.119 | 0.925±0.000 | 0.961±0.000 | 0.150±0.000 |

Table 7. Ablation study of Ex4DGS on the Flame Salmon from the N3DV dataset.

| SSU | RSR | $\epsilon$ | SAA | Opacity Regularization | PSNR | SSIM | MS-SSIM | LPIPS(vgg) |
|---|---|---|---|---|---|---|---|---|
| x | x | $10^{-8}$ | x | x | 28.719±0.152 | 0.925±0.001 | 0.955±0.001 | 0.140±0.003 |
| ✓ | x | $\epsilon_e(10^{-7})$ | x | x | 29.163±0.039 | 0.927±0.000 | 0.957±0.001 | 0.136±0.000 |
| ✓ | ✓ | $\epsilon_e(10^{-7})$ | x | x | 29.358±0.308 | 0.928±0.000 | 0.957±0.001 | 0.140±0.003 |
| ✓ | ✓ | $\epsilon_e(10^{-7})$ | x | ✓ | 29.403±0.300 | 0.928±0.001 | 0.957±0.000 | 0.138±0.000 |
| ✓ | ✓ | $\epsilon_e(10^{-7})$ | ✓ | ✓ | 29.398±0.380 | 0.928±0.001 | 0.957±0.001 | 0.138±0.000 |

## F.2. Cut Roasted Beef from N3DV

Table 8. Ablation study of 4DGS on the Cut Roasted Beef from the N3DV dataset.

| Batch Size | SSU | RSR | $\epsilon$ | SAA | PSNR | SSIM | MS-SSIM | LPIPS(vgg) |
|---|---|---|---|---|---|---|---|---|
| 4 | x | x | $10^{-15}$ | x | $32.239 \pm 1.091$ | $0.952 \pm 0.000$ | $0.977 \pm 0.001$ | $0.139 \pm 0.004$ |
| 4 | ✓ | ✓ | $\epsilon_e\ (10^{-8})$ | x | $34.202 \pm 0.246$ | $0.958 \pm 0.000$ | $0.980 \pm 0.000$ | $0.136 \pm 0.000$ |
| 4 | ✓ | ✓ | $\epsilon_e\ (10^{-8})$ | ✓ | $34.321 \pm 0.256$ | $0.960 \pm 0.000$ | $0.981 \pm 0.000$ | $0.135 \pm 0.000$ |
| 1 | ✓ | ✓ | $\epsilon_e\ (10^{-8})$ | ✓ | $33.617 \pm 0.231$ | $0.958 \pm 0.000$ | $0.979 \pm 0.000$ | $0.145 \pm 0.000$ |

Table 9. Ablation study of Ex4DGS on the Cut Roasted Beef from the N3DV dataset.

| SSU | RSR | $\epsilon$ | SAA | Opacity Regularization | PSNR | SSIM | MS-SSIM | LPIPS(vgg) |
|---|---|---|---|---|---|---|---|---|
| x | x | $10^{-8}$ | x | x | $32.310 \pm 0.369$ | $0.956 \pm 0.000$ | $0.975 \pm 0.000$ | $0.132 \pm 0.001$ |
| ✓ | x | $10^{-6}$ | x | x | $33.013 \pm 0.531$ | $0.961 \pm 0.000$ | $0.967 \pm 0.128$ | $0.129 \pm 0.000$ |
| ✓ | x | $\epsilon_e:$ $(\epsilon_s = 10^{-6}, \epsilon_d = 10^{-7})$ | x | x | $32.853 \pm 1.363$ | $0.962 \pm 0.000$ | $0.978 \pm 0.000$ | $0.126 \pm 0.001$ |
| ✓ | ✓ | $\epsilon_e:$ $(\epsilon_s = 10^{-6}, \epsilon_d = 10^{-7})$ | x | x | $33.539 \pm 0.308$ | $0.962 \pm 0.000$ | $0.978 \pm 0.000$ | $0.125 \pm 0.000$ |
| ✓ | ✓ | $\epsilon_e:$ $(\epsilon_s = 10^{-6}, \epsilon_d = 10^{-7})$ | ✓ | x | $33.612 \pm 0.273$ | $0.963 \pm 0.000$ | $0.978 \pm 0.000$ | $0.125 \pm 0.000$ |
| ✓ | ✓ | $\epsilon_e:$ $(\epsilon_s = 10^{-6}, \epsilon_d = 10^{-7})$ | x | ✓ | $33.550 \pm 0.467$ | $0.963 \pm 0.000$ | $0.979 \pm 0.000$ | $0.125 \pm 0.000$ |
| ✓ | ✓ | $\epsilon_e:$ $(\epsilon_s = 10^{-6}, \epsilon_d = 10^{-7})$ | ✓ | ✓ | $33.727 \pm 0.326$ | $0.963 \pm 0.000$ | $0.979 \pm 0.000$ | $0.125 \pm 0.000$ |

## F.3. Coffee Martini from N3DV

Table 10. Ablation study of 4DGS on the Coffee Martini from the N3DV dataset.

| Batch Size | SSU | RSR | $\epsilon$ | SAA | PSNR | SSIM | MS-SSIM | LPIPS(vgg) |
|---|---|---|---|---|---|---|---|---|
| 4 | x | x | $10^{-15}$ | x | $28.279 \pm 0.035$ | $0.917 \pm 0.000$ | $0.954 \pm 0.000$ | $0.151 \pm 0.000$ |
| 4 | ✓ | ✓ | $\epsilon_e\ (10^{-8})$ | x | $29.051 \pm 0.062$ | $0.920 \pm 0.000$ | $0.958 \pm 0.000$ | $0.149 \pm 0.000$ |
| 4 | ✓ | ✓ | $\epsilon_e\ (10^{-8})$ | ✓ | $29.125 \pm 0.036$ | $0.922 \pm 0.000$ | $0.959 \pm 0.000$ | $0.148 \pm 0.000$ |
| 1 | ✓ | ✓ | $\epsilon_e\ (10^{-8})$ | ✓ | $28.942 \pm 0.157$ | $0.919 \pm 0.000$ | $0.958 \pm 0.000$ | $0.159 \pm 0.000$ |

Table 11. Ablation study of Ex4DGS on the Coffee Martini from the N3DV dataset.

| SSU | RSR | $\epsilon$ | SAA | Opacity Regularization | PSNR | SSIM | MS-SSIM | LPIPS(vgg) |
|---|---|---|---|---|---|---|---|---|
| x | x | $10^{-8}$ | x | x | $28.356 \pm 0.226$ | $0.920 \pm 0.001$ | $0.952 \pm 0.001$ | $0.151 \pm 0.004$ |
| ✓ | x | $\epsilon_e(10^{-7})$ | x | x | $28.856 \pm 0.221$ | $0.923 \pm 0.001$ | $0.953 \pm 0.002$ | $0.148 \pm 0.000$ |
| ✓ | ✓ | $\epsilon_e(10^{-7})$ | x | x | $28.664 \pm 0.193$ | $0.921 \pm 0.000$ | $0.951 \pm 0.000$ | $0.149 \pm 0.000$ |
| ✓ | ✓ | $\epsilon_e(10^{-8})$ | x | ✓ | $28.953 \pm 0.185$ | $0.923 \pm 0.001$ | $0.955 \pm 0.002$ | $0.146 \pm 0.001$ |
| ✓ | ✓ | $\epsilon_e(10^{-8})$ | ✓ | ✓ | $28.930 \pm 0.197$ | $0.924 \pm 0.001$ | $0.955 \pm 0.002$ | $0.146 \pm 0.001$ |

## F.4. Cook Spinach from N3DV

Table 12. Ablation study of 4DGS on the Cook Spinach from the N3DV dataset.

| Batch Size | SSU | RSR | $\epsilon$ | SAA | PSNR | SSIM | MS-SSIM | LPIPS(vgg) |
|---|---|---|---|---|---|---|---|---|
| 4 | x | x | $10^{-15}$ | x | 33.018±0.399 | 0.955±0.001 | 0.977±0.000 | 0.138±0.000 |
| 4 | ✓ | ✓ | $\epsilon_e$ ($10^{-8}$) | x | 33.488±0.093 | 0.956±0.000 | 0.979±0.000 | 0.138±0.000 |
| 4 | ✓ | ✓ | $\epsilon_e$ ($10^{-8}$) | ✓ | 33.704±0.066 | 0.958±0.000 | 0.979±0.000 | 0.137±0.000 |
| 1 | ✓ | ✓ | $\epsilon_e$ ($10^{-8}$) | ✓ | 33.300±0.032 | 0.955±0.000 | 0.977±0.000 | 0.147±0.000 |

Table 13. Ablation study of Ex4DGS on the Cook Spinach from the N3DV dataset.

| SSU | RSR | $\epsilon$ | SAA | Opacity Regularization | PSNR | SSIM | MS-SSIM | LPIPS(vgg) |
|---|---|---|---|---|---|---|---|---|
| x | x | $10^{-8}$ | x | x | 32.332±0.333 | 0.956±0.000 | 0.974±0.000 | 0.130±0.002 |
| ✓ | x | $10^{-6}$ | x | x | 32.734±0.338 | 0.957±0.000 | 0.973±0.000 | 0.131±0.000 |
| ✓ | x | $\epsilon_e$ : ($\epsilon_s = 10^{-6}, \epsilon_d = 10^{-7}$) | x | x | 32.574±0.422 | 0.959±0.000 | 0.974±0.000 | 0.129±0.000 |
| ✓ | ✓ | $\epsilon_e$ : ($\epsilon_s = 10^{-6}, \epsilon_d = 10^{-7}$) | x | x | 33.025±0.107 | 0.959±0.000 | 0.975±0.000 | 0.128±0.000 |
| ✓ | ✓ | $\epsilon_e$ : ($\epsilon_s = 10^{-6}, \epsilon_d = 10^{-7}$) | ✓ | x | 32.998±0.128 | 0.960±0.000 | 0.975±0.000 | 0.128±0.000 |
| ✓ | ✓ | $\epsilon_e$ : ($\epsilon_s = 10^{-6}, \epsilon_d = 10^{-7}$) | x | ✓ | 32.896±0.116 | 0.959±0.000 | 0.975±0.000 | 0.128±0.000 |
| ✓ | ✓ | $\epsilon_e$ : ($\epsilon_s = 10^{-6}, \epsilon_d = 10^{-7}$) | ✓ | ✓ | 32.928±0.111 | 0.960±0.000 | 0.975±0.000 | 0.128±0.000 |

## F.5. Flame Steak from N3DV

Table 14. Ablation study of 4DGS on the Flame Steak from the N3DV dataset.

| Batch Size | SSU | RSR | $\epsilon$ | SAA | PSNR | SSIM | MS-SSIM | LPIPS(vgg) |
|---|---|---|---|---|---|---|---|---|
| 4 | x | x | $10^{-15}$ | x | 28.358±0.595 | 0.954±0.001 | 0.968±0.002 | 0.134±0.000 |
| 1 | ✓ | ✓ | $\epsilon_e$ ($10^{-8}$) | x | 33.802 ± 0.116 | 0.961 ± 0.000 | 0.980± 0.000 | 0.135 ±0.000 |
| 1 | ✓ | ✓ | $\epsilon_e$ ($10^{-8}$) | ✓ | 33.951 ± 0.943 | 0.963 ± 0.000 | 0.981 ± 0.000 | 0.134 ± 0.000 |

Table 15. Ablation study of Ex4DGS on the Flame Steak from the N3DV dataset.

| SSU | RSR | $\epsilon$ | SAA | Opacity Regularization | PSNR | SSIM | MS-SSIM | LPIPS(vgg) |
|---|---|---|---|---|---|---|---|---|
| x | x | $10^{-8}$ | x | x | 33.138±0.515 | 0.962±0.008 | 0.977±0.001 | 0.120±0.003 |
| ✓ | x | $10^{-6}$ | x | x | 33.372±0.336 | 0.966±0.000 | 0.979±0.001 | 0.118±0.000 |
| ✓ | x | $\epsilon_e$ : ($\epsilon_s = 10^{-6}, \epsilon_d = 10^{-7}$) | x | x | 33.346±0.886 | 0.966±0.000 | 0.980±0.001 | 0.115±0.000 |
| ✓ | ✓ | $\epsilon_e$ : ($\epsilon_s = 10^{-6}, \epsilon_d = 10^{-7}$) | x | x | 34.041±0.196 | 0.967±0.000 | 0.980±0.000 | 0.114±0.000 |
| ✓ | ✓ | $\epsilon_e$ : ($\epsilon_s = 10^{-6}, \epsilon_d = 10^{-7}$) | ✓ | x | 34.015±0.186 | 0.967±0.000 | 0.981±0.000 | 0.114±0.000 |
| ✓ | ✓ | $\epsilon_e$ : ($\epsilon_s = 10^{-6}, \epsilon_d = 10^{-7}$) | x | ✓ | 34.027±0.036 | 0.967±0.000 | 0.980±0.000 | 0.115±0.000 |
| ✓ | ✓ | $\epsilon_e$ : ($\epsilon_s = 10^{-6}, \epsilon_d = 10^{-7}$) | ✓ | ✓ | 34.032±0.059 | 0.967±0.000 | 0.980±0.000 | 0.114±0.000 |

## F.6. Sear Steak from N3DV

*Table 16.* Ablation study of 4DGS on the Sear Steak from the N3DV dataset.

| Batch Size | SSU | RSR | $\epsilon$ | SAA | PSNR | SSIM | MS-SSIM | LPIPS(vgg) |
|---|---|---|---|---|---|---|---|---|
| 4 | x | x | $10^{-15}$ | x | 30.329±1.438 | 0.956±0.005 | 0.973±0.006 | 0.133±0.002 |
| 1 | ✓ | ✓ | $\epsilon_e$ $(10^{-8})$ | x | 33.607 ± 0.088 | 0.961 ± 0.000 | 0.980 ± 0.000 | 0.137 ± 0.000 |
| 1 | ✓ | ✓ | $\epsilon_e$ $(10^{-8})$ | ✓ | 33.819 ±0.123 | 0.963 ± 0.000 | 0.982 ± 0.000 | 0.136 ± 0.000 |

*Table 17.* Ablation study of Ex4DGS on the Sear Steak from the N3DV dataset.

| SSU | RSR | $\epsilon$ | SAA | Opacity Regularization | PSNR | SSIM | MS-SSIM | LPIPS(vgg) |
|---|---|---|---|---|---|---|---|---|
| x | x | $10^{-8}$ | x | x | 33.037±0.435 | 0.960±0.000 | 0.977±0.002 | 0.119±0.002 |
| ✓ | x | $10^{-6}$ | x | x | 33.660±0.19 | 0.967±0.000 | 0.981±0.000 | 0.117±0.000 |
| ✓ | x | $\epsilon_e$ : $(\epsilon_s = 10^{-6}, \epsilon_d = 10^{-7})$ | x | x | 33.583±0.735 | 0.968±0.000 | 0.981±0.000 | 0.114±0.000 |
| ✓ | ✓ | $\epsilon_e$ : $(\epsilon_s = 10^{-6}, \epsilon_d = 10^{-7})$ | x | x | 34.075±0.368 | 0.968±0.000 | 0.981±0.001 | 0.113±0.000 |
| ✓ | ✓ | $\epsilon_e$ : $(\epsilon_s = 10^{-6}, \epsilon_d = 10^{-7})$ | ✓ | x | 34.197±0.381 | 0.968±0.000 | 0.982±0.000 | 0.112±0.000 |
| ✓ | ✓ | $\epsilon_e$ : $(\epsilon_s = 10^{-6}, \epsilon_d = 10^{-7})$ | x | ✓ | 34.127±0.300 | 0.968±0.000 | 0.982±0.000 | 0.114±0.000 |
| ✓ | ✓ | $\epsilon_e$ : $(\epsilon_s = 10^{-6}, \epsilon_d = 10^{-7})$ | ✓ | ✓ | 34.212±0.274 | 0.968±0.000 | 0.982±0.000 | 0.114±0.000 |

