# OpenReview forum: "Improving Explicit Dynamic Gaussian Splatting Optimization via Update Mixture"
_ICML.cc/2026/Conference — ICML 2026 regular_

### Official Review · Reviewer_22ZV · 2026-03-09

**Soundness:** 4
**Presentation:** 2
**Significance:** 3
**Originality:** 3
**Overall Recommendation:** 4
**Confidence:** 3

**Summary:**

The authors address the overfitting issue in GS by employing a Strictly Sparse Update strategy and incorporating regularization terms. By appropriately selecting the $\epsilon$ term in the Adam optimizer, they enable a smoother transition adaptive to non-adaptive updates. And a parameter averaging method is adopted to mitigate the tendency of primitives to develop preferences for specific frames.

**Compliance With Llm Reviewing Policy:**

Affirmed.

**Final Justification:**

The response has  adddress  my concerns, the score is adjusted accordingly.

**Key Questions For Authors:**

See weaknesses part.

**Limitations:**

Yes

**Strengths And Weaknesses:**

Strengths
1 The methods proposed in the paper are relatively easy to implement and demonstrate promising performance in experiments, indicating significant practical value.

2 The article features comprehensive citations, with every proposed method supported by relevant literature to clarify the origin of the underlying ideas.

3 The study includes rigorous ablation experiments that provide substantial evidence confirming the effectiveness of the proposed approach.

Weaknesses
1 The article frequently introduces methods by directly citing literature (e.g., RSR and SWA) without providing brief descriptions. The readers need to conduct extensive searches and memorize concepts while reading, making the paper less accessible.

2  In Section 4.1, the regularization terms are only described textually, making it difficult to clearly understand the exact loss function definition. It would be better to include precise mathematical formulations.

3 For Figure 2, it would be preferable to use a logarithmic scale rather than a uniform scale for the y-axis. In Ex4DGS, $\sqrt{v_0}$ is very close to zero, and without a logarithmic scale, it's hard to verify whether "$\sqrt{v}$ rapidly stabilizes at a fixed order of magnitude" as claimed in Section 4.2.

4 Section 4.3 states that SAA introduces almost no additional memory overhead. However, it requires at least additional storage space equal to the parameter size to store the averaged values. I don't believe this memory overhead can be considered negligible.

---

> ### Author Rebuttal · Authors · 2026-03-30
>
> Thanks for this comment, below are the responses.
>
> The anonymous **link**: https://anonymous.4open.science/r/3DGS_UMR4_ICML-4BBB/README.md
>
> **For Weaknesses 1 & 2**:
> - ReState Regularization (RSR): Given the adaptive first moment $m_k$ and second moment $v_k$, we consider using RSR to attenuate the memory effect. Specifically, at a predefined iteration interval, we sample a subset of primitives and reset the corresponding $m_k$ and $v_k$ for the sampled primitives. Concretely, we obtain updated moments $m_k^{\text{new}}$ and $v_k^{\text{new}}$ via  Eq.1. $\alpha_1^2=\alpha_2$ is enforced to preserve the magnitude of the update step. We adopt the default setting from the original paper, namely $\alpha_1=0.2$ and $\alpha_2=0.04$. Under this mechanism, the sampled primitives are continuously subjected to a memory penalty. (We will provide further details in the appendix.) $$
> m_k^{\text{new}}=\alpha_1 m_k,\qquad v_k^{\text{new}}=\alpha_2 v_k,  \tag{1}
> $$
> - SWA：At present, we have already described the formulation of SAA and its specific usage in the main paper. later, we will further include the implementation details of SWA in the appendix.
> - Regularization for Ex4DGS: In the Method section, we state that, in our implementation, the motion regularization adopted in Ex4DGS is removed. In Appendix B.2, we provide a detailed discussion of the key implementation procedures of Ex4DGS, with particular emphasis on the definitions related to motion. Furthermore, in Appendix C.3, we present an explicit formulation of the motion regularization term, based on which we further analyze its gradient scale and explain in detail why we ultimately discard the current motion regularization strategy.
> $$\nabla \mathcal{R}(o) =  \min(\lambda_o \frac{\nabla o / N_{I}}{\sqrt{\hat{v}(o)_{t}^{\prime}}+\epsilon}, \mathrm{Const})] \tag{2}$$
> - Opacity regularization: We employ a **decoupled opacity regularization** in our method. Specifically, on top of the original optimizer update, we compute the magnitude of the opacity regularization according to Eq. 2. The resulting term does not participate in the original gradient computation; instead, it is directly added to the update step produced by the optimizer, after which the parameters are updated according to Eq. 3. We use the default hyperparameter settings. (Further details will be included in the appendix.)
>
> $$\theta(o)_{k+1}=\theta(o)-\eta \times [\frac{\hat{m}(o)_k}{\sqrt{\hat{v}(o)_k}+\epsilon} + \nabla R(o)]\tag{3}$$ where $N_I$ denotes the image scale, and $\theta(o)$, $\hat{m}(o)$, and $\hat{v}(o)$ denote the opacity parameter and its associated optimizer states, namely the bias-corrected first and second moments, respectively. Here, $\lambda_o$ is the corresponding hyperparameter.
>
> **For Weaknesses 3:** We will supplement the corresponding images in the anonymous link (see Fig.1 in the link). Thank you for your suggestion.
>
> **For Weaknesses 4:** SAA only requires averaging the parameters from designated checkpoints, and these averaged parameters are not involved in subsequent training. Accordingly, there are at least two possible implementation strategies. The first is to maintain the running average online on the GPU, which admittedly requires additional storage proportional to the parameter size. The second is to avoid maintaining averaged parameters during training, and instead save the corresponding parameters as checkpoints and perform SAA after training. In this case, no additional GPU memory is required. We will include a further discussion of SAA implementations in the appendix.

---

> > ### Author Rebuttal · Reviewer_22ZV · 2026-04-05
> >
> > First, I would like to thank the authors for their detailed responses to my questions. Regarding w.1 and w.2, the authors have provided fairly comprehensive explanations about the specific implementation details of the method. Adding these implementation details will significantly improve the paper's readability. The logarithmic-scale plot the authors provided in response to w.3 clearly supports their statements in Section 4.2.
> >
> > However, concerning w.4, the statement in the paper that "Following SWA, the additional computation and memory overhead is negligible" remains inappropriate. I suggest the authors modify this statement and include in the appendix a discussion of the exact computational overhead (what percentage this step's computation time accounts for in the total computation time) and VRAM overhead (if using sliding average on GPU) that different implementation approaches would introduce. This would help readers choose which specific solution to adopt.
> >    Based the response, I'll adjust the score.

---

> > > ### Author Response · Authors · 2026-04-06
> > >
> > > We thank the reviewer for the valuable suggestion. We will revise the statement in the main text and add an appendix discussion providing the relevant technical details, including quantitative comparisons of the computational and VRAM overhead under different implementation choices.

---

### Official Review · Reviewer_Y2SX · 2026-03-12

**Soundness:** 3
**Presentation:** 3
**Significance:** 2
**Originality:** 2
**Overall Recommendation:** 4
**Confidence:** 2

**Summary:**

The paper addresses a real and practically issue in explicit Dynamic Gaussian Splatting: novel-view performance can deteriorate late in training, especially in large-motion scenes. It proposes a modular update-mixture approach and shows meaningful gains on both 4DGS and Ex4DGS.

**Compliance With Llm Reviewing Policy:**

Affirmed.

**Final Justification:**

The reviewer has addressed most of my concerns.

**Key Questions For Authors:**

1. The motivation for SAA is frame preference, but the evidence seems to rely mainly on test-set fluctuation curves and modest PSNR gains, while the paper also says SAA does not consistently improve all Ex4DGS/N3DV scenes. Could the authors provide more direct evidence that frame preference is the causal issue—for example, per-frame variance analyses, temporal-consistency metrics, or comparisons against simpler averaging schemes?
2. For 4DGS, some of the strongest improvements appear together with disabling batch sampling, and Table 4 even mixes batch-size settings across scenes. Could the authors disentangle how much of the final gain comes from the proposed optimizer changes versus the change from batch size 4 to 1?
3. The paper is framed as an optimization study of explicit Dynamic GS, but experiments are limited to two baselines and two datasets. How broadly do the authors expect the conclusions to transfer to other dynamic GS pipelines, especially those with different motion parameterizations or stronger geometric priors?

**Limitations:**

yes

**Strengths And Weaknesses:**

# Strengths
1. The paper identifies a real and important failure mode in explicit Dynamic GS: test-view quality can drop late in training, especially in large-motion scenes, and it shows this consistently on both 4DGS and Ex4DGS.
2. The method is easy to understand and ablate because each component targets a specific issue: SSU for accessible updates, epsilon correction for over-adaptivity, and SAA for frame preference
3. The reported results show meaningful gains on both main baselines, with improvements in average PSNR and supporting qualitative examples that suggest the gains are visually relevant rather than metric-only

# Weakness
1. The paper’s three components look more like a tailored combination of existing optimizer ideas than a fundamentally new optimization principle.
2. The paper explicitly says the method may fail when the learning rate is not properly tuned and that a well-calibrated learning rate is essential. Table 7 also shows that changing the learning-rate configuration alone can alter Ex4DGS performance, which weakens the claim that the gains come cleanly from the proposed method rather than improved optimization settings.

---

> ### Author Rebuttal · Authors · 2026-03-31
>
> Thanks for this comment.
> The anonymous link: https://anonymous.4open.science/r/3DGS_UMR3_ICML-40D2/README.md
>
> For **W1**:  1.Our method is motivated by understanding of both the optimizer and GS, and is driven by a well-defined problem rather than a simple combination of existing techniques. In practice, it effectively addresses the issue and achieves consistent improvements. 2.Current evidence suggests that **Adam** remains a more suitable optimizer for GS (we have also tested alternatives such as Lion and SWATS). We therefore believe that further refining Adam is both necessary and meaningful.
>
> For **W2**:(1) On **Technicolor**, **Ex4DGS** uses notably different hyperparameters across scenes. (2) We revisit the **opacity-related hyperparameters** because, as discussed in **Sec. 4, Scale of Gradients (ii&iii)**, opacity affects both **scale** and **spatial relationships**. The two ablated opacity learning rates differ by **500$\times$**; since our method directly targets the **adaptive update**, such a gap can naturally invalidate it. (3) Our ablation shows that, under the smaller setting, performance does not drop, and our method still brings further gains. We therefore believe this comparison further highlights the effect of the **update step** in Dynamic GS, consistent with our central discussion. (4) We will include this analysis in the appendix.
>
> For **Q1**:
> - The motivation for frame preference: 1. As shown by the PSNR curves in Fig. 4 in paper, we decompose the training-time PSNR drop into two factors: generalization and **frame preference**. Here, **frame preference** manifests as a sudden drop followed by a recovery in PSNR during training, indicating that the model temporarily learns representations that favor training but harm testing. This behavior is potentially correctable. Since our goal is to achieve the best possible result at the end of training, we introduce SAA to mitigate this effect. For example, on Cook Spinach, the test PSNR is 33.64 at 29k iters but drops to 33.55 at 30k; with SAA, it improves to 33.75.
> - 2.We observe that SAA can improve certain fine-grained reconstruction details, and we provide a subset of the corresponding visual evidence in the anonymous link Fig1.
> - We will add this discussion to the paper.
>
> For **Q2**:
> - The original **4DGS** cannot be used with **batch sizw(B)=1**, as already stated in the original paper. We also provide corresponding visual evidence in **Fig. 3** of our paper, showing that it produces severe artifacts under **B=1**.
> - With our optimization, **4DGS** no longer produces such artifacts (**see Fig. 3**).
> - We therefore did not report the original **4DGS** with **$B=1$** in the paper, since the original method is problematic in this setting. After enabling **$B=1$** with our method, the training time can be substantially reduced under the same number of iterations (about **50%** in our experiments).
> - We discuss **batch sampling** separately in the paper, together with the presence of **mixed batch-size settings**, because we find that although batch sampling is widely used in Dynamic GS, it may itself introduce instability.
> - In **Appendix C.4**, we discuss how **batch sampling** can lead to an **underestimation of the moments in Adam**, while also suppressing part of the effective gradients. In the anonymous link Fig.2, we further provide **primitive-wise optimization info** for **$B=4$**, together with corresponding samples for the **same primitive group** under **$B=1$**. Under batch sampling, the primitives exhibit **continuous small-amplitude oscillations**, along with occasional abrupt changes. In contrast, when **$B=1$**, the gradient information is **sparser**, while the larger gradient scale may provide the constraint on the **2nd moment**.
> - Our experiments on **4DGaussian** show that **$B=1$** yields better results (**see Q3**). Therefore, our work discusses that **batch sampling may introduce risks**, which is why we provide **4DGS without batch sampling**.
>
> For **Q3**: To further validate the **generalization** of our method, we additionally extend it to **4DGaussian**. Unlike the methods studied in our paper, **4DGaussian**[3] introduces an additional **MLP** to Gaussian primitives to model deformation and temporal information, resulting in a more complex optimization system. Nevertheless, since Gaussian primitives remain a core component, the issue discussed in our paper still persists in this setting. We directly extend our method to this case and further apply **$\epsilon_e$** to constrain the optimization of the **MLP**. Both the experimental results and the corresponding ablation studies show that our method remains effective, and that **$\epsilon_e$** is also beneficial for the MLP optimization. We provide a more detailed technical analysis in our response to reviewer `7dCk`, **S3&S4**. The results are directly available in the anonymous link Tab1-3.
>
> [3] 4D Gaussian Splatting for Real-Time Dynamic Scene Rendering

---

> > ### Author Rebuttal · Reviewer_Y2SX · 2026-04-07
> >
> > I would like to thank the authors for addressing the most of my concerns.

---

### Official Review · Reviewer_7dCk · 2026-03-12

**Soundness:** 2
**Presentation:** 2
**Significance:** 3
**Originality:** 3
**Overall Recommendation:** 4
**Confidence:** 4

**Summary:**

This paper studies the generalization degradation problem in explicit Dynamic Gaussian Splatting (DynamicGS), especially in scenes with large motion. The authors analyze the optimization behavior of DynamicGS and identify two key issues: non-uniform gradient scaling caused by adaptive optimizers and frame preference introduced by the frame-by-frame optimization process. To address these problems, the paper proposes an update mixture strategy that combines adaptive and non-adaptive updates through a constant-corrected adaptive algorithm. In addition, the method introduces Strictly Sparse Update (SSU) to stabilize updates and Stochastic Attribute Averaging (SAA) to mitigate frame preference.

**Compliance With Llm Reviewing Policy:**

Affirmed.

**Final Justification:**

The authors have addressed the concerns raised in my review during the rebuttal phase, and I have increased my score accordingly.

**Key Questions For Authors:**

### Suggestions for Improvement

1. Conduct additional experiments on more recent dynamic Gaussian-based reconstruction methods to verify whether the observed generalization degradation is a common issue.

2. Include statistical analysis or visualizations of primitive-wise gradient distributions across frames to empirically support the claim that gradient variability contributes to optimization instability.

3. Provide a clearer discussion or additional experiments to illustrate the scenarios in which the proposed update mixture strategy is effective, such as whether it generalizes to other explicit dynamic Gaussian pipelines beyond the evaluated methods.

4. Applying the proposed update mixture strategy to more recent dynamic Gaussian reconstruction methods would help demonstrate whether the approach remains effective beyond the two evaluated pipelines.

**Limitations:**

yes

**Strengths And Weaknesses:**

### Summary of Strengths

1. **Clear problem analysis.**
   The paper identifies a generalization degradation issue in explicit Dynamic Gaussian Splatting and provides a clear analysis linking it to adaptive optimization and frame preference.

2. **Reasonable optimization strategy.**
   The proposed update mixture strategy combines adaptive and non-adaptive updates, along with SSU and SAA, to stabilize optimization and improve generalization.

### Summary of Weaknesses

1. The observed generalization degradation is mainly demonstrated on two explicit DynamicGS pipelines, namely 4DGS and Ex4DGS. It remains unclear whether this issue consistently appears in other dynamic Gaussian-based methods. Additional experiments on more DynamicGS variants or recent approaches would help verify the generality of this observation and rule out potential pipeline-specific effects.

2. The paper suggests that the gradients of the same Gaussian primitive may vary significantly across different frames, which may contribute to optimization instability. However, the current discussion is largely qualitative. Providing quantitative statistics or visualizations of gradient distributions across frames would offer stronger empirical support for this claim.

3. The paper primarily evaluates the proposed strategy on explicit DynamicGS pipelines without MLP-based temporal modeling, but the broader applicability of the update mixture strategy is not clearly discussed. A clearer discussion of the applicable scenarios, such as whether the method generalizes to other explicit dynamic Gaussian methods beyond the tested pipelines, would improve the clarity of the contribution.

---

> ### Author Rebuttal · Authors · 2026-03-31
>
> Thanks for this comment. As the Weaknesses largely overlap with the Suggestions, we address the Suggestions directly below.
> The anonymous **Link**: https://anonymous.4open.science/r/3DGS_UMR2_ICML-589B/README.md
>
> For **S1**: We additionally provide the example of Swift4D[1] in the link (Fig.1c), which suggests that **generalization issue is a common issue in Dynamic GS**. We also provide additional evidence (Fig.1a&b) in 4DGaussian[2]. For 4DGaussian, we repeated the experiments multiple times across different hardware platforms and consistently observed similar behavior.
>
> For **S2**: We provide the **primitive-wise optimizer information** in the anonymous link (Fig.2) to support our claim. These samples are collected from the opacity-related optimizer states of primitives during training iterations 16.5k–17.0k, from which the variation of quantities such as the gradients can be directly observed.
>
> For **S3&S4**:
> To the best of our knowledge, the currently open-sourced **explicit dynamic GS** are primarily **4DGS** and **Ex4DGS**, which also represent two distinct technical paradigms. To further substantiate the **generalization** of our method, we will additionally extend our methods to **4DGaussian**[2], which represents another fundamentally different implementation of dynamic GS. We will further discuss its relation to our method(P1&P2), the validity of the supporting evidence(P2), the extension of our approach to this setting(P3), the resulting performance(P4) and the conclusions:
> - **P1**: Relation between the two paradigms.: We first clarify the relation between methods represented by **4DGaussian**, which incorporate an **implicit structure (MLP)** and which we refer to below as **Semi-Explicit Dynamic GS**, and the Explicit Dynamic GS discussed in our paper. Explicit Dynamic GS reconstructs the scene solely through Gaussian primitives defined under different formulations, where the temporal dimension is typically modeled **explicitly**. This makes it possible to analyze the **gradient flow** in Explicit Dynamic Gaussian Splatting in a relatively clean manner, focusing directly on the gradients associated with the primitives themselves. By contrast, Semi-Explicit Dynamic GS introduces an additional MLP to Gaussian primitives to model temporal information. Under this setting, the optimization exhibits both the gradient characteristics of Gaussian primitives and the optimization behavior of neural networks. As discussed in the Introduction of our paper, this leads to **a more complex optimization system**. Therefore:
> - **P2**:  (1) By starting from **Explicit Dynamic GS**, our discussion more directly captures the optimization limitations of Gaussian primitives in Dynamic GS. (2) **Semi-Explicit Dynamic GS** inherits both the optimization deficiencies of Gaussian primitives and the potential risks introduced by the MLP. For example, Eq. 5 in our paper indicates an inherent **gradient-scale imbalance** induced by Gaussian primitives, which may further lead to degraded neural representation capacity [3]. (3) _This hierarchical relation also implies that our method can be naturally extended to **Semi-Explicit Dynamic GS** methods represented by **4DGaussian**._
> - **P3**. Extending our method to 4DGaussian: We evaluate our method on four highly dynamic scenes from **N3DV**, namely Cut Roasted Beef, Cook Spinach, Flame Steak & Sear Steak. The overall setup follows the configuration used in our paper. In all experiments, we use batch size = 1, since, as analyzed in our paper, batch sampling may also introduce potential instability to adaptive updates. For the **coarse training stage** in 4DGaussian (which only includes **3k iterations**), we keep the original $\epsilon$ unchanged. For the **fine training stage**, we replace $\epsilon_e$ with our method. In addition, **we further conduct an ablation study to examine whether $\epsilon_e$ should also be applied to the neural network components**.
> - **P4**. results(Link Tab.1-3): After further extending $\epsilon_e$ to the optimization of the MLP components, **our method also improves the stability of 4DGaussian, a Semi-Explicit Dynamic GS method**. This further demonstrates the effectiveness. For the primitive, we keep the same $\epsilon_e$ estimation strategy as in our method. For the MLP part, using **100$\times \epsilon_e$** yields more stable results on **Flame Steak**, while **$\epsilon_e$** is kept unchanged for the other scenes.
> - **P5**: The above results show that, (1), **our method exhibits strong generality and can be readily extended to Semi-Explicit Dynamic GS**; (2) they also reveal a **broadly existing issue in current Dynamic GS** and suggest **an effective direction for addressing it**.
>
> [1] Swift4D:Adaptive divide-and-conquer Gaussian Splatting for compact and efficient reconstruction of dynamic scene. ICLR 2025
>
> [2] 4D Gaussian Splatting for Real-Time Dynamic Scene Rendering
>
> [3] Gradient starvation: A learning proclivity in neural networks

---

> > ### Author Rebuttal · Reviewer_7dCk · 2026-04-03
> >
> > The authors have largely addressed my concerns through additional experiments.

---

### Official Review · Reviewer_czSW · 2026-03-12

**Soundness:** 3
**Presentation:** 3
**Significance:** 2
**Originality:** 3
**Overall Recommendation:** 4
**Confidence:** 2

**Summary:**

I am not deeply familiar with the optimization aspects of this domain, so I kindly ask the Area Chair to weigh this review alongside those of other reviewers who may have stronger expertise in this area.

The paper introduces an "update mixture" strategy aimed at alleviating the generalization degradation commonly observed in explicit Dynamic Gaussian Splatting pipelines, especially in scenes exhibiting large inter-frame motions.

**Compliance With Llm Reviewing Policy:**

Affirmed.

**Final Justification:**

Many thanks to the author, most of my concerns have been addressed

**Key Questions For Authors:**

- How generalizable is the proposed method across different Dynamic GS pipelines? Does it require method-specific adaptations and hyperparameter tuning for each underlying base approach, or can it be applied in a more plug-and-play manner?
- What scale of inter-frame motion qualifies as "large motion" in this context? How robust is the proposed method to variations in the magnitude of motion (e.g., from moderate to extremely large displacements)? Additional analysis or experiments characterizing robustness with respect to motion scale would be valuable.

**Limitations:**

yes

**Strengths And Weaknesses:**

#### Strengths

- The proposed "update mixture" strategy provides meaningful and substantial help in improving generalization performance in dynamic scenes.
- Figures 3 and 4, together with Tables 1 and 2, clearly and effectively illustrate the improvements and behavioral changes brought by incorporating the proposed method.
- The work offers a solid theoretical analysis of the underlying causes of poor generalization, followed by a suite of targeted algorithmic contributions: (i) Strictly Sparse Update (SSU) enforced via temporal accessibility masks together with additional regularizations on opacity and ReState; (ii) constant-corrected adaptive steps (using a large ϵ_e) that naturally blend adaptive and non-adaptive update behaviors; and (iii) Stochastic Attribute Averaging (SAA).

#### Weaknesses

- While the method yields noticeable improvements on the training set, the gains on the test set—as reported in the tables—are comparatively modest and limited in magnitude.
- The experimental evaluation remains somewhat narrow. Stronger evidence would come from results across a broader range of scenes and additional datasets.
- The estimation of ϵ_e still appears to require manual, per-method and per-scene calibration, which reduces the practicality and automation of the approach.

---

> ### Author Rebuttal · Authors · 2026-03-31
>
> Thanks for this comment, below are the responses. (W: Weaknesses; Q: Questions)
>
> The anonymous **Link**: https://anonymous.4open.science/r/3DGS_UM_ICML-5E07/README.md
>
> For **W1**: All the results reported in our paper are evaluated on the **test set**. Without changing the original pipeline, our method improves **PSNR by approximately 2 dB in Table 4** and by **about 1 dB in Table 3**. Such gains are widely regarded as meaningful in the literature we cited. More importantly, our method provides a **more general improvement in training stability**.
>
> For **W3**:
> - For the estimation of **$\epsilon$**, we provide an initial strategy that uses the average 2nd moment at the initial steps as a reference. This approach is applicable to the vast majority of cases. For the few scenarios requiring additional adjustment, such as **Flame Salmon** and **Coffee Martini** in **Ex4DGS**, we have provided the related analysis in the paper. For **scenes with small motion**, a more relaxed **$\epsilon$** can be adopted, since under such conditions the adaptive learning rate is generally more reliable.
> - Using the reference estimate we provide still yields a value higher than the original one. We provide an example of **Flame Salmon** in the anonymous link.
> - In the paper, the adjustment of **$\epsilon$** serves two purposes: 1), it directly improves the **reconstruction quality**; 2), it provides the most direct **evidence** of the potential risk of adaptive learning rates in **Dynamic GS**, while also suggesting a practical direction for addressing this issue.
> - Our method remains effective when extended to Semi-Explicit Dynamic GS (see below Q1&W2).
>
> For **Q1 & W2**：
> To further validate the **generalization** of our method, we provide two additional experiments: (1) extending our method to 4DGaussian[1], and (2) applying it to Ex4DGS on MeetRoom[2].
> - about experiment (1) in Link Tab.1-3:
> - - The main experiments in the current paper focus on **Explicit Dynamic GS**, which reconstructs scenes solely with Gaussian primitives. In contrast, methods represented by **4DGaussian** can be viewed as introducing an **implicit structure (MLP)** to Gaussian primitives to model temporal or deformation information; in the following, we refer to this line as **Semi-Explicit Dynamic GS**. **4DGaussian** represents a different technical route for realizing Dynamic GS. We provide a more detailed discussion of the relation between **Explicit and Semi-Explicit Dynamic GS** in our response to reviewer `7dCk`, **S3&S4, P1&P2**. Because Gaussian primitives remain a core component, **Semi-Explicit Dynamic GS** also inherits the issue discussed in our paper. Extending our method to **4DGaussian** therefore provides further evidence for the **generalization** of our approach.
> - - We provide a more detailed description of the extension procedure in our response to reviewer `7dCk`, **S3&S4, P3**. Briefly, following the exploration in our paper, we set the **batch size=1**. During the **coarse training** (which only includes 3k iterations), we keep **$\epsilon$** unchanged. During the **fine training**, we apply our estimation strategy to set **$\epsilon_e$** for the primitive parameters. We further extend **$\epsilon_e$** to the **MLP** optimization, and our ablation study shows that it is necessary. Except for **Flame Steak**, where **100$\times\epsilon_e$** is used for the MLP part, **$\epsilon_e$** is kept unchanged in all other cases.
> - - Our method can be effectively applied to **4DGaussian**, demonstrating strong **generalization** and promising **plug-and-play** potential. At the same time, it further reveals a **widely existing issue in current Dynamic GS **, including the optimization of **MLP part**, and suggests an effective direction for addressing it.
> - about experiment (2) in Link Tab.4-6:
> - - MeetRoom satisfies the same conditions as the datasets discussed in our paper, namely **real-world, multi-view, and long-sequence**. Our data processing pipeline and parameter settings follow those used for **N3DV**.
> - - Ex4DGS achieves a clear improvement on MeetRoom.
>
> For **Q2**：
> Our method is fundamentally aimed at improving **optimization stability**. It does not introduce additional geometric cues such as **optical flow** or **depth**, and therefore cannot fundamentally compensate for the geometric limitations of the underlying method itself. We discuss its performance limitation under **extremely large displacements** in **Appendix C.5** and **Fig 10**. We believe that **optimization stability is also a fundamental research issue**, and we will further investigate these extreme cases in future work.
>
> [1] 4D Gaussian Splatting for Real-Time Dynamic Scene Rendering
>
> [2] Streaming radiance fields for 3d video synthesis.

---

### Decision · Program_Chairs · 2026-04-30

**Decision:**

Accept (regular)

**Comment:**

This paper receives 4x weak accepts. All reviewers think that the paper has identified a generalization degradation issue in explicit Dynamic Gaussian Splatting, and the proposed "update mixture" strategy provides meaningful and substantial help in improving generalization performance in dynamic scenes.  The work offers a solid theoretical analysis of the underlying causes of poor generalization, followed by a suite of targeted algorithmic contributions: i) Strictly Sparse Update (SSU) ii) constant-corrected adaptive steps and iii) Stochastic Attribute Averaging (SAA). Furthermore, the methods proposed in the paper are relatively easy to implement and demonstrate promising performance in experiments, indicating significant practical value. The AC follows the suggestions of the reviewers to accept the paper.